# Testing hypothetical bias in a choice experiment: An application to the value of the carbon footprint of mandarin oranges

**Keiko Aoki** [1]*, **Kenju Akai** [2]

**1** Platform of Inter/Transdisciplinary Energy Research, Kyushu University, Fukuoka, Japan, **2** Center for the Promotion of Project Research and Center for Community-based Health Research and Education, Shimane University, Izumo, Shimane, Japan

* aoki.keiko.729@m.kyushu-u.ac.jp, fuizu43@gmail.com

**Data Availability Statement:** All relevant data are within the paper and its Supporting Information files.

**Funding:** The work was supported by Grants-in-Aid for Scientific Research #23730276 (Recipient:

## Abstract

This study investigates "hypothetical bias," defined as the difference in the willingness to pay for a product attribute between hypothetical and non-hypothetical conditions in a choice experiment, for the carbon footprint of mandarin oranges in Japan. We conducted the following four treatments: a non-hypothetical lab economic experiment, a hypothetical lab survey, a hypothetical online survey, and a hypothetical online survey with cheap-talk. Each treatment asked participants to choose one of three oranges based on price and carbon emissions level. Next, participants were asked to answer questions on demographics and the following three kinds of environmental factors: environmental consciousness, purchasing behavior for goods with eco-labels, and daily environmental behavior. Using the random parameter logit model, the willingness to pay per 1g of carbon emission reduction were 0.53 JPY, 0.52 JPY, 0.54 JPY, and 0.58 JPY in the non-hypothetical lab economic experiment, hypothetical lab survey, hypothetical online survey and hypothetical online survey with cheap-talk, respectively. The complete combinatorial test of the willingness to pay for carbon emission reductions indicates no hypothetical bias between any treatment combinations. Our findings reveal that environmental attributes for food are less likely to show hypothetical bias than other goods. The results of the main effect with an interaction term show that environmental consciousness reduces the coefficients of carbon emissions in all treatments. Therefore, a psychological scale is useful for showing whether hypothetical bias emerges with treatment or participants' personal backgrounds.

## 1 Introduction

The choice experiment (CE) is a type of stated preference method [1] that elicits consumer preferences based on various scenarios, unlike the revealed preference method. In a CE, participants choose one of several alternatives based on different levels of attributes. To measure consumers' willingness to pay (WTP), the CE is a useful method for examining how consumers choose products and services, such as food and beverages, transportation, environment, and health [1,2].

Keiko Aoki, Ph.D.), the Japan Society for the Promotion of Science.

**Competing interests:** The authors have declared that no competing interests exist.

The CE is often employed as part of a survey in which participants do not buy goods or services; this method is therefore called a "hypothetical CE." Conversely, a "non-hypothetical CE" includes participants with monetary incentives applied through the experimental economics method developed by Vernon Smith based on the principles of the induced value theory [3], in which participants are awarded cash for their performance in an experiment and cannot be deceived. Thus, in a non-hypothetical CE, the experimenter provides real goods, including the attributes they offer, and participants receive money equal to the endowment minus the price of the goods they chose in the CE [4,5].

Comparing hypothetical and non-hypothetical CE methods reveals hypothetical bias (HB) [6]. Here, HB means that WTP for an attribute in a non-hypothetical CE is lower than that in a hypothetical CE. This concept was originally discussed in relation to the contingent valuation method (CVM) [7–9]: the inclusion of cheap-talk scripts in the CVM [10] can make participants more cautious and prevent HB. Therefore, this CE study considers HB to be present if the WTP for an attribute in the hypothetical CE with cheap-talk scripts is lower than that in the hypothetical CE.

Table 1 summarizes previous studies on HB in the context of food by employing CE. Notably, Lusk and Schroeder were the first to study HB. In the case of beef steak, they found HB in total WTP, but not in marginal WTP [6]. Carlsson, Frykblom, and Lagerkvist found HB in WTP for genetically modified organisms (GMO) in chicken and beef [11]. Chang, Lusk, and Norwood observed HB toward organic beef and wheat; however, they forecasted market share and did not estimate WTP [12]. Similarly, Yue and Tong showed HB toward organic and local tomatoes [13]. Also notable is that Aoki, Shen, and Saijo revealed HB toward sodium nitrite in ham sandwiches [4]. Additionally, De-Magistris, Gracia, and Nayga Jr. found HB toward organic food and food miles in the case of almonds, but this HB disappeared when cheap-talk scripts and their original statements were included [14]. Grebitus, Lusk, and Nayga found HB toward apples and wine as well as food miles with regard to total WTP, conditional on participants' personal traits; however, they did not show marginal WTP [15]. Moser, Raffaelli, and Notaro found no HB toward the reduced impact of the climate on apples [16]. Alemu and Olsen, employing customized cheap-talk scripts and an augmented opt-out option, found HB toward cricket powder in bread [17]. Liebe et al. found HB toward organic and fair trade in tea in online environments [18]. Last, Wuepper, Clemm, and Wree found HB toward water footprint labels in coffee with regard to the marginal effect in online shops [19].

In addition to these studies in food, several CE studies reveal HB in environmental projects. Carlsson and Martinsson, the pioneers of testing HB in CE, did not find HB in this context [20]. Meanwhile, Araña and León investigated the test dynamics of HB in environmental projects over several months, and found that while HB was observed in the first month, it disappeared in the second month [21]. Johansson-Stenman and Svedsäter found HB for WWF donations but not local restaurant vouchers [22]. Hensher observed no HB for travel route choices [23]. As for studies on HB in other fields, Haghani et al. summarized [24,25].

Governments often employ online surveys to investigate how food labeling and packaging relate to consumer perceptions of value. While online surveys are useful and convenient, they are often criticized as involving HB. Furthermore, laboratory economic experiments have become increasingly popular but are also criticized for using small samples that do not reflect real markets and only recruit participants from the locality surrounding the laboratory, unlike online surveys, which can have a national scope. Therefore, experimental results have been criticized for their poor applicability to policy implications. Comparing laboratory economic experiments with online surveys reveals a tradeoff between a small number of samples with monetary incentives and a large number of samples without monetary incentives. However, based on Table 1, no study using CE has yet compared whether laboratory economic experiments or online surveys are more effective for testing new food policies.

**Table 1. Summary of previous studies on testing hypothetical bias by employing food.**

| Authors | Environments (#of subject) | | Treatments (n = # of subject) | Goods | Attributes or labels | Hypothetical bias (Marginal WTP): |
|---|---|---|---|---|---|---|
| | Control (= Hypothetical CE) (n = # of subject) | Mode and location of incentive compatible setting | | | | H0:WTP(t)-WTP(c) = 0 H1:WTP(t)-WTP(c)<0 |
| Lusk and Schroder (2004) | Laboratory on the university (n = 37) Reward: fixed money | Laboratory on the university Reward: money = endowment-binding price | Non-hypothetical CE (n = 67) | Beef | Generic Guaranteed tender Natural Choice Certified Angus Beef | No No No No No |
| Carlsson et al. (2005) | Mail survey | Mail survey | Hypothetical CE with cheap-talk | Chicken | Growth GMO Out summer Mobile | Yes Yes Yes Yes |
| | | | Hypothetical CE with cheap-talk | Beef | Improved labelling GMO Out all year Mobile | No Yes No No |
| Chang et al. (2009) [a] | Laboratory on the university (n = 47) Reward: fixed money | Laboratory on the university Reward: money = endowment-binding price | Non-hypothetical CE (n = 46) | Ground beef | Fresh Lean Diet Lean Organic | N.A. N.A. N.A. N.A. |
| | | | Non-hypothetical CE (n = 46) | Wheat flour | Hodgson Mill King Arthur Gold Medal GO Organic | N.A. N.A. N.A. N.A. |
| Yue and Tong (2009) | Minnesota State Fair (n = 233) Reward: small money or gift | Minnesota State Fair Reward: small money or gift | Non-hypothetical CE (n = 110) | Tomato | Organic Local | Yes Yes |
| Aoki et al. (2010) | Street survey in Osaka (n = 445) Reward: canned juice | Laboratory on the university: Reward: money = total (endowment-price) | Non-hypothetical CE (n = 117) | Ham sandwich | Sodium Nitrite | Yes |
| De-Magistris et al. (2013) | Lab setting in a town of Spain (n = N.A.) | Lab setting in a town of Spain Reward: money = endowment-binding price | Non-hypothetical CE (n = N.A.) | Almond | EU organic label Food miles labels | Yes Yes |
| | | | Hypothetical CE with cheap-talk (n = N.A.) | Almond | EU organic label Food miles labels | Yes No |
| | | | Neutral priming in the hypothetical CE (n = N.A.) | Almond | EU organic label Food miles labels | No No |
| | | | Neutral priming in the non-hypothetical CE (n = N.A.) | Almond | EU organic label Food miles labels | N.A. N.A. |
| | | | Honesty priming in the hypothetical CE (n = N.A.) | Almond | EU organic label Food miles labels | Yes Partially Yes |
| | | | Honesty priming in the non-hypothetical CE (n = N.A.) | Almond | EU organic label Food miles labels | N.A. N.A. |
| Grebitus et al. (2013) [b] | Lab setting in Bonn and Cologne(n = 48) Reward: fixed money | Lab setting in Bonn and Cologne: Reward: money = endowment-binding price | Non-hypothetical CE (n = 50) | Apple | Food miles: | N.A. WTP considered personal trait |
| | | | Non-hypothetical CE (n = 50) | Wine | Food miles: | N.A. |

*(Continued)*

**Table 1.** (Continued)

| Authors | Environments (#of subject) | | Treatments (n = # of subject) | Goods | Attributes or labels | Hypothetical bias (Marginal WTP): |
|---|---|---|---|---|---|---|
| | Control (= Hypothetical CE) (n = # of subject) | Mode and location of incentive compatible setting | | | | H0:WTP(t)-WTP (c) = 0 H1:WTP(t)-WTP (c)<0 |
| Moser et al. (2013) | Interview at the super in Trentino(n = 96) Reward: dried apple slices | Interview at the super in Trentino Reward: dried apple slices Pay with participants' own money | Real CE (n = 96) | Apple | Method Appearance Origin Reduced impact on climate | Yes (Organic)/ No (IPM, Innovative) Yes Yes No |
| | | | Hypothetical CE with cheap-talk (n = 96) | Apple | Method Appearance Origin Reduced impact on climate | No Yes (Mediocre) / No (Good) No No |
| Alemu and Olsen (2018) | Interview in Kenya (n = 109) Reward: fixed money | Interview in Kenya Reward: money = endowment-binding price | Non-hypothetical CE (n = 116) | Bread added edible insects | Amount of cricket flour Whether some portion of the wheat flour is fortified or not | Yes Yes |
| | | | Hypothetical CE with CT augmented an Opt-out reminder (n = 109) | Bread added edible insects | Amount of cricket flour Whether some portion of the wheat flour is fortified or not | Yes Yes |
| Liebe et al. (2019) | Online (n = 157) Reward: fixed money | Online Reward: money = endowment-binding price | Real CE (n = 142) | Tea | Organic Fair trade | Yes Yes |
| Wuepper et al. (2019) [c] | Store and cafe survey in Munich (n = 759) Reward: no | Real online shop Reward: no, paying with participants' own money | Non-hypothetical CE (n = 693) | Coffee | Intensity Origin Organic Water footprint Label Color of package | N.A. N.A. N.A. N.A. N.A. N.A. |

Notes

[a], [b] and [c] do not estimate WTP for each attribute in the studies.

This background prompted us to employ CE to investigate the existence of HB by comparing laboratory economic experiments with online surveys. We conducted four treatments: a non-hypothetical lab economic experiment (NHLEE), a hypothetical lab survey (HLS), a hypothetical online survey (HOS), and a hypothetical online survey with cheap-talk (HOSCT).

We used Japanese mandarin oranges (*Citrus unshiu* Marc.) as the good, which are very similar to the mandarin oranges popular in Europe and America (*Citrus reticulata*) and one of the most popular fruits in Japan. We selected the oranges' carbon footprint as the attribute. The Paris Climate Conference (COP21) recently spotlighted Climate Action, one of the 17 United Nations' sustainable development goals (SDGs), which promotes reducing carbon footprints while simultaneously strengthening food security, including food production [26]. Governments are encouraged to evaluate the carbon footprints of their products and invest in products that can mitigate climate change [27]. In Japan, transport emissions account for approximately 80% of all domestic emissions; thus, the Japanese government must reduce transportation-related carbon emissions [28]. The fact that large metropolitan areas must

import crops from remote regions is responsible for a good deal of carbon emissions in Japan. However, to date, the value of carbon footprint labeling to consumers, which is not legally required, is unknown. To work toward the SDGs, the value of carbon footprint labelling for consumers should be clarified [29].

Although the Japanese government has been interested in carbon footprints since 2008, the topic has received less and less attention since 2013 and is rarely discussed today [30]. In a pioneering hypothetical survey CE for Japanese consumer research, Kimura et al. found that a carbon footprint label induces Japanese consumers' WTP [31]. Along these same lines, Aoki and Akai investigated the value of the carbon footprint of Japanese oranges by comparing general environmental scales [32] in a hypothetical survey. They showed that consumers with high environmental consciousness valued carbon footprints [33]. However, Aoki and Akai's economic experiment to investigate the value of carbon footprints for oranges found no significant difference in WTP from that in Aoki & Akai [33], which indicates that no HB was present [34]. Outside of Japan, Macdiarmid et al.'s non-hypothetical choice experiment found that consumers were willing to pay a premium for lasagna with a lower carbon footprint [35].

No studies have yet used experimental economics to investigate HB toward carbon footprints. Responding to this lack, this study offers insights useful for government decisions about whether to introduce a carbon footprint label. Notably, several studies have used hypothetical survey CEs to measure WTP for a reduced carbon footprint. Pioneers Onozaka and McFadden reported that carbon footprint labels for apples and tomatoes are more valued when combined with other sustainable labels, such as organic and fair trade labels [36]. Caputo, Nayga Jr, and Scarpa found that consumers valued carbon footprint labels more than food miles labels for fresh tomatoes [37]. Additionally, Grebitus, Steiner, and Veeman investigated the value of the carbon footprint compared to the water footprint for ground beef by comparing the impact of consumers' human value systems on food choices [38], and showed that a personal propensity to purchase influences the value of the carbon footprint [39]. Van Loo, Caputo, Nayga Jr, and Verbeke found that high-income earners valued carbon footprint labels more than other sustainable labels, such as organic, animal welfare, and free-range labels on chicken breasts [40]. The results of Thøgersen and Nielsen implied that people prefer the low carbon footprint and that the carbon footprint increases people's environmental consciousness [41]. Apostolidis and McLeay reported that sustainability-oriented consumers prefer low carbon footprint labels to other sustainability labels for mincemeat [42]. Moreover, De Marchi et al. showed the effects of attaching importance to the carbon footprint [43]. Lombardi et al. found that communication influenced the value of the carbon free products [44]. Emberger-Klein and Menrad and Feucht and Zander investigated the design of the carbon footprint [45,46].

Since these studies found that other eco-labels and environmental awareness affected preference for CFP, this study also analyzed whether personal attributes, environmental psychology, and environmental consideration scales affected preference for CFP to examine the social factors that promote CFP.

The paper is organized as follows. Sections 2 and 3 explain the material and methods, and model, respectively. Section 4 describes the results. Section 5 discusses the results and Section 6 concludes the paper.

## 2 Material and methods

### 2.1 Experimental design

This study compares four treatments as shown in Table 2. First, the NHLEE employed the experimental economics method and was conducted in the university laboratory. Participants

**Table 2. The characteristics of treatments.**

| | Non-hypothetical Lab Economic Experiment (NHLEE) | Hypothetical Lab Survey (HLS) | Hypothetical Online Survey (HOS) | Hypothetical Online Survey with Cheap-talk (HOSCT) |
|---|---|---|---|---|
| Place | Lab | Lab | Online | Online |
| Cheap-talk | No | No | No | Yes |
| Oranges | Real | Photo | Photo | Photo |
| Monetary incentive | Yes | No | No | No |
| # of respondents | 104 | 212 | 500 | 500 |

received an endowment to buy real, fresh oranges. They choose one of three oranges packed in a clear plastic bag. Second, the HLS employed a survey questionnaire conducted in the university laboratory. Participants were asked to imagine they had an endowment to buy real, fresh oranges. They chose one of the three oranges photographed in the NHLEE.

Third, the HOS employed an online survey questionnaire conducted by Rakuten Insight Global, which has the largest single panel in Japan. Participants were asked to imagine they had an endowment to buy real, fresh oranges. They chose one of the three oranges photographed in the NHLEE. Fourth, the HOSCT was the same as the HOS except that it included a cheap-talk script inspired by Carlsson et al. [11]. The cheap-talk script is provided in the Supplementary Materials.

For the online survey, the company recruited the respondents, and managed the participant identities. The authors did not know any personal identification for any respondents. The data collection complied with the Law on the Protection of Personal Information in Japan. The institutions and universities to which the authors belong do not require ethical approval for science research, except in instances that could be deemed life-threatening or harmful to human subjects. Each respondent participated in only one session.

For this study, we conducted the HOS and HOSCT treatments and used the date from the NHLEE and the HLS conducted in Aoki and Akai [34] and Aoki and Akai [33], respectively.

NHLEE and HLS were conducted in 2012 and HOS and HOSCT were conducted in 2016. During the last eight years, CFP was not popularized, and no laws were made about it. The Consumer Price Index (CPI) [47] for 2012 and 2016 were 97.2 and 99.9, respectively, with 2015 being 100. 2012–2016 was a sluggish period in Japan's prices, as it was below 100, and the situation was similar for the previous eight years. In addition, the inflation rate announced by the IMF [48] was -0.06% in 2012 and -0.12% in 2016, both of which were implemented during a period of decline. Thus, Japan's economic situation has been implemented at a similar time.

## 2.2 Products and carbon footprints

As noted in the Introduction, we used the Japanese orange, satsuma mandarin (*Citrus unshiu* Marc.) as our product. The Japanese orange is one of the most popular fruits in Japan—it is produced in 36 out of all 46 prefectures [49] and is ranked third in terms of share in total fruit consumption [50], with almost 100% production self-sufficiency in Japan since 1965 [51]. All products were selected from three of the top five origins, namely, the Wakayama, Ehime, and Kumamoto prefectures, located 100km, 380km, and 800km respectively from the university laboratory in NHLEE.

The carbon footprint was calculated based on the four stages of the life cycle assessment: production [52,53], sorting and packing [54], transportation [55], and packaging [56]. In the production stage, the level of carbon emissions mainly depends on whether the product is cultivated in a greenhouse or outdoors. This study used oranges grown outdoors and transported

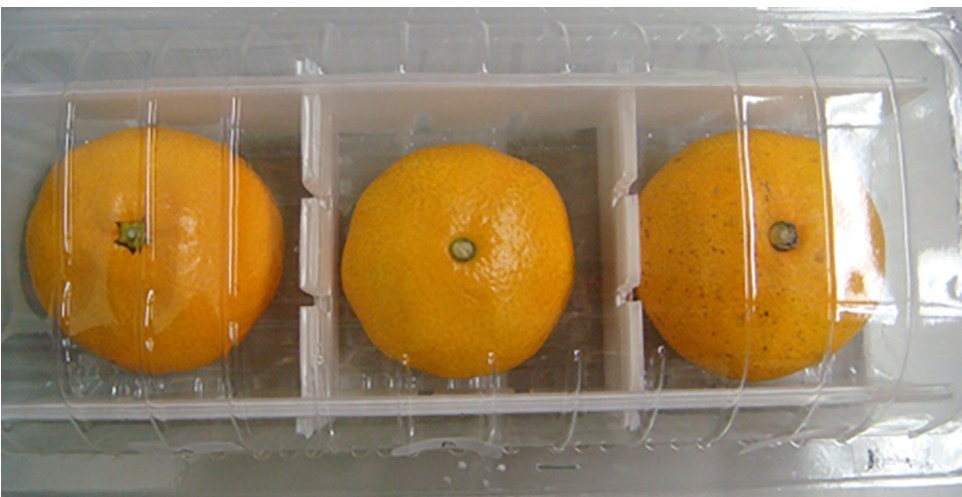

**Fig 1. A picture of the oranges used in three treatments as the HLS, the HOS, and the HOSCT.**

by truck because it was conducted while they were in season. Among the four stages, the largest proportion of carbon emissions is generated during transportation.

Since the orange-growing process is the same across all three prefectures, the main difference in the carbon footprint of the oranges arises from the distance factor: the carbon emissions of each orange produced in Wakayama, Ehime, and Kumamoto are 20g, 30g, and 40g, respectively. We used these emission levels to measure the oranges' carbon footprint attribute. Participants were informed that the differences in the magnitude of the carbon footprint were primarily due to the distance from the origin. However, unlike in the case of food miles, the carbon footprint is calculated in terms of $CO_2$; therefore, participants did not know the origin of each orange.

We select oranges of size S, which are, on average, 7cm in diameter and 100g in weight. These oranges are relatively small, and smaller oranges are generally known to be sweeter than larger ones. Participants were informed about the size of the oranges; in fact, they could see the real oranges in the NHLEE and, in the other treatments, the photographs included their sizes.

Fig 1 shows the three oranges used in the NHLEE treatment.

## 2.3 Choice set design

We used three Japanese orange alternatives, A, B, and C, and two attributes, price and carbon emissions (as a measure of the carbon footprint). The price had three levels per orange: 25 JPY, 35 JPY, and 45 JPY. These levels are based on two sources: the average retail price of oranges in the three largest supermarkets near the university laboratory and the average national price in all other supermarkets and shops in Japan [57]. The price levels across these sources are not significantly different from each other. The carbon emissions had three levels per orange, as advised in Section 2.2: 20g, 30g, and 40g per orange.

To create choice sets based on these price and carbon emission levels, we employed a D-optimal fractional factorial design using Design Expert 7.0, which is more useful for creating a reasonable number of choice sets in practice compared to a full factorial design. Ultimately, we created 24 choice sets and divided them into two blocks of 12 per the below reasoning.

Design Expert 7.0 (State Easy) was used to design the choice set, with D-optimal as the design type and the coordinate-exchange algorithm (CEA) as the efficient search algorithm. CEA is known to work very well under multinomial logit discrete choice models [58].

|  | Orange A | Orange B | Orange C |
|---|---|---|---|
| Price | 35 JPY | 25 JPY | 45 JPY |
| Carbon dioxide emissions | 30 g | 20 g | 40g |
| I would choose… | ☐ | ☐ | ☐ |

**Fig 2. An example of a choice set.**

In stores near our campus, size S oranges are usually sold in packs of 6. Given the 12 choice sets, two packs of oranges were used, totaling 12 oranges with a weight of approximately 1.2kg. The actual monthly consumption of oranges when they are in season (between November and March) is approximately 3kg per person. Thus, the weight of the 12 oranges in the study, 1.2kg, is equivalent to 40% of actual monthly household consumption. Because too many oranges indicate a heuristic participant decision, we ensured the amounts indicated reasonable decisions. Fig 2 illustrates an example of the designated choice sets.

Finally, we did not provide an opt-out option. While previous studies indicate that an opt-out option increases the realism of choice by preventing forced choices [1,23], Lusk and Schroeder and Alemu and Olsen found that the non-hypothetical condition induced a higher rate of choosing the opt-out option than the hypothetical condition does[6,17]. Notably, Carlsson, Frykblom, and Lagerkvist found that a CE with an opt-out option has greater unobserved heterogeneity than one without it [59]. We assumed that the induced value theory in experimental economics does not work in non-hypothetical experiments when an opt-out option is introduced. In the experiment, if we had introduced an opt-out option, then participants may have a different purpose from the experimenter. This is because the objective of some participants changes from choosing food, the purpose of the experiment, to just making money. Because the main premise of experimental economics is based on the induced value theory it cannot be a valid basis for the experiments in this study.

## 2.4 Questionnaire

After the CE, we conduct questionnaires consisting of five parts, as shown in Table 3. The first part comprised demographics such as gender, age, household, education, and income. The

**Table 3. Demographics.**

| Variable | Definition | Non-hypothetical Lab Economic Experiment (NHLEE), | Hypothetical Lab Survey (HLS) | Hypothetical Online Survey (HOS) | Hypothetical Online Survey with Cheap-talk (HOSCT) | All sample |
|---|---|---|---|---|---|---|
| Gender | Male | 34.6% | 40.1% | 59.0% | 56.6% | 46.9% |
|  | Female | 65.4% | 59.9% | 41.0% | 43.4% | 53.1% |
| Age | 18–19 years old | 2.9% | 34.4% | 6.0% | 6.0% | 10.3% |
|  | 20–24 years old | 33.7% | 18.4% | 3.8% | 4.0% | 8.6% |
|  | 25–29 years old | 5.8% | 1.4% | 10.2% | 10.0% | 8.4% |
|  | 30–34 years old | 4.8% | 3.3% | 7.0% | 8.6% | 6.8% |
|  | 35–39 years old | 11.5% | 3.3% | 11.8% | 10.2% | 9.8% |
|  | More than 40 years old | 41.3% | 39.2% | 61.2% | 61.2% | 56.1% |

*(Continued)*

**Table 3.** (Continued)

| Variable | Definition | Non-hypothetical Lab Economic Experiment (NHLEE), | Hypothetical Lab Survey (HLS) | Hypothetical Online Survey (HOS) | Hypothetical Online Survey with Cheap-talk (HOSCT) | All sample |
|---|---|---|---|---|---|---|
| Household | 1 person | 25.0% | 31.1% | 11.2% | 14.2% | 16.6% |
| | 2 persons | 10.6% | 17.0% | 28.2% | 28.4% | 25.1% |
| | 3 persons | 18.3% | 18.9% | 27.2% | 26.4% | 24.8% |
| | 4 persons | 30.8% | 23.1% | 21.0% | 18.6% | 21.2% |
| | 5 persons | 13.5% | 6.6% | 8.6% | 9.8% | 9.1% |
| | More than 6 persons | 1.0% | 3.3% | 3.8% | 2.6% | 3.0% |
| Education | High school | 7.7% | 42.5% | 31.4% | 30.2% | 30.9% |
| | Academy | 4.8% | 5.7% | 8.8% | 11.6% | 9.0% |
| | Community college or university | 72.1% | 49.5% | 50.8% | 49.6% | 51.8% |
| | Graduate school | 13.5% | 0.9% | 6.0% | 6.4% | 5.9% |
| | Others | 0.0% | 1.4% | 3.0% | 2.2% | 2.2% |
| Annual Income | Less than 2.5 million JPY | 21.2% | 33.5% | 11.2% | 14.6% | 16.9% |
| | 2.5–4.0 million JPY | 4.8% | 14.2% | 20.8% | 21.8% | 18.8% |
| | 4.0–5.5 million JPY | 8.7% | 7.5% | 19.0% | 21.6% | 17.3% |
| | 5.5–7.0 million JPY | 14.4% | 12.7% | 16.8% | 15.2% | 15.3% |
| | More than 7.0 million JPY. | 34.6% | 19.3% | 32.2% | 26.8% | 28.3% |
| ECCB | Average total scale (S.D.) | 30.42 (6.79) | 28.80 (7.74) | 27.56 (8.69) | 28.06 (8.41) | 54.3% |
| The frequency of eating the oranges (Variable: Frequency) | 1: Once or less than once a week | 21.2% | 26.9% | 65.4% | 65.4% | 55.7% |
| | 2: A few times a week | 56.7% | 62.7% | 23.6% | 22.8% | 32.2% |
| | 3: Almost every day | 22.1% | 10.4% | 11.0% | 11.8% | 12.1% |
| Do you buy goods attached eco-label? [a] (Variable: Label) | 1:Yes; 0:No | 97.1% | 97.2% | 93.4% | 95.8% | 95.2% |
| Eco behavior in daily life [a] | I have not used any shopping bags when I buy something: 0:Yes; 1:No (Variable: Not using shopping bags) | 66.0% | 62.7% | 63.0% | 64.6% | 63.8% |
| | I often use public transportation or a bicycle, but not a car: 1:Yes; 0:No (Variable: Public Transportation) | 50.5% | 43.9% | 22.0% | 21.4% | 27.5% |
| | I do not leave the tap running and often turn off a light when not in use: 0:Yes; 1:No (Variable: Not tap running) | 89.3% | 84.4% | 68.0% | 71.8% | 73.8% |
| | I adjust the room temperature in accordance with health: 1:Yes; 0:No (Variable: Air Conditioning) | 78.6% | 73.6% | 53.4% | 52.4% | 58.3% |
| | I always use the stairs and not elevators and escalators: 1:Yes; 0:No (Variable: Walking) | 25.2% | 29.7% | 22.0% | 20.4% | 22.9% |
| | I separate the garbage: 1:Yes; 0:No (Variable: Garbage) | 86.4% | 76.9% | 69.2% | 70.2% | 72.2% |
| | Other eco behaviors apart from above six questions (free-writing) [b] | 13.6% | 8.5% | 7.2% | 4.4% | 6.8% |
| # of respondents | | 104 | 212 | 500 | 500 | 1316 |

Notes: Standard errors are in parentheses.

[a] is a dummy variable in the model.

[b] is not used in the model.

**Table 4. Average ECCB scale.**

| No | Syntax of the scale | Non-hypothetical Lab Economic Experiment (NHLEE), | Hypothetical Lab Survey (HLS) | Hypothetical Online Survey (HOS) | Hypothetical Online Survey with Cheap-talk (HOSCT) |
|---|---|---|---|---|---|
| 1 | I have purchased a household appliance because it uses less electricity than other brands. | 3.44 (1.09) | 3.41 (1.04) | 3.07(1.05) | 3.16 (1.03) |
| 2 | I have purchased light bulbs that are more expensive but save energy. | 2.77 (1.23) | 2.47 (1.23) | 2.82 (1.16) | 2.85 (1.17) |
| 3 | I will not buy products that have excessive packaging. | 2.95 (1.16) | 2.81 (1.17) | 2.80 (1.09) | 2.90 (1.11) |
| 4 | If I understand the potential damage to the environment that some products can cause, I do not purchase these products. | 3.60 (0.95) | 3.43 (1.18) | 2.90 (1.13) | 2.96 (1.05) |
| 5 | I have switched products for ecological reasons. | 2.93 (0.93) | 2.79 (1.04) | 2.61 (1.12) | 2.69 (1.06) |
| 6 | I have convinced members of my family or friends not to buy some products that are harmful to the environment. | 1.98 (0.93) | 1.87 (1.08) | 2.20 (1.13) | 2.17 (1.13) |
| 7 | Whenever possible, I buy products packaged in reusable containers. | 3.08 (0.96) | 2.78 (1.17) | 2.74 (1.09) | 2.78 (1.06) |
| 8 | When I have a choice between two equal products, I always purchase the one that is less harmful to other people and the environment. | 3.54 (1.07) | 3.36 (1.16) | 2.78 (1.11) | 2.84 (1.10) |
| 9 | I will not buy a product if the company that sells it is ecologically irresponsible. | 3.17 (1.09) | 2.87 (1.20) | 2.81 (1.16) | 2.85 (1.11) |
| 10 | I do not buy household products that harm the environment. | 2.96 (1.16) | 3.06 (1.18) | 2.84 (1.19) | 2.85 (1.11) |
| | Total | 30.42 (6.79) | 28.80 (7.74) | 27.56 (8.69) | 28.06 (8.41) |
| | Cronbach's Alpha | 0.81 | 0.88 | 0.93 | 0.91 |

Notes: Standard deviations are in parentheses. Scoring scale reversed the order: Always true = 5, mostly true = 4, sometimes true = 3, rarely true = 2, and never true = 1.

second part comprised the ecologically conscious consumer behavior (ECCB) scale [32], a psychological scale measuring environmental purchase behavior. We employed 10 items following Johnston, Wessells, Donath, & Asche [60], extracted from the 30 original items. The 10 items are summarized in Table 4. The responses were rated on a 5-point Likert-type scale, ranging from 1 ("always agree") to 5 ("never agree"). The third part examined the frequency of eating oranges per week. The fourth part comprised the purchasing behavior in relation to goods with an eco-label, developed specifically for this study, to test whether consumers prefer goods with an eco-label. The fifth part examined the daily environmental behavior, developed specifically for this study. This part consisted of six questions to test whether participants practice environmentally beneficial behavior in their daily lives. We employ these questions in the estimation model to further investigate the preferences regarding price and carbon footprint.

## 2.5 Procedures

**2.5.1 NHLEE.** We conducted the NHLEE treatment in the laboratory at Osaka University, Japan. The NHLEE treatment consisted of the following seven steps:

*Step 1*: Each participant sat behind a desk, separated by 60cm × 80cm partitions in front and beside the desk, in the university laboratory, as shown in Fig 3.

*Step 2*: Each participant submitted a signed consent form before the experiment began.

*Step 3*: Each participant received an instructional leaflet explaining the experiment's procedure; the experimenter also read the instructions aloud once.

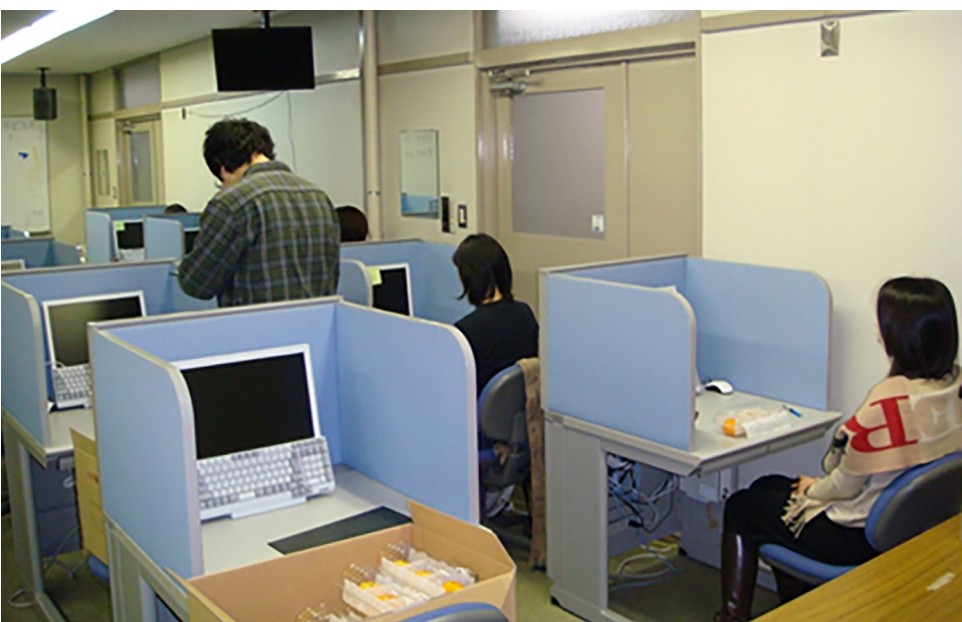

**Fig 3. A picture in the lab.**

*Step 4*: Each participant received a choice set card and a clear plastic-hinged box containing three oranges, each with a different carbon emission level, as shown in Figs 1 and 2.

*Step 5*: Each participant selected one of the three oranges to buy with their endowment of 120 JPY. Each participant then wrote down the orange chosen in the choice set card. Next, the experimenters gathered the cards and the box. These processes constituted one round. We conducted 12 rounds.

*Step 6*: Each participant filled in the questionnaire after completing all rounds.

*Step 7*: Each participant received the rewards and selected oranges from all rounds. The rewards were calculated by the following equation: show-up fee and the sum of the changes of the endowment minus the price of the selected orange in each round. After a 10% income tax was deducted from the rewards, the remaining money was handed to the participants.

The entire process lasted for approximately 60 minutes. The instructions and questionnaires are presented in the Supplementary Materials.

**2.5.2 HLS.** We conduct the HLS treatment in the laboratory at Osaka University, Japan. The HLS treatment follows the same steps as the NHLEE, except for Steps 4, 5, and 7. In Step 4, each participant received a choice set card and a photograph of the oranges taken in the NHLEE, as mentioned in Section 2.1. In Step 5, the experimenters gathered only the cards. In Step 7, each participant received a fixed reward of 1,000 JPY. The process lasted for approximately 45 minutes.

**2.5.3 HOS.** The HOS treatment involved an online survey with the following five steps:

*Step 1*: Each participant read an online instruction that explained the experiment's procedure.

*Step 2*: Each participant viewed an online choice set card and photographs of the oranges taken in the NHLEE.

*Step 3*: Each participant imagined that they had an endowment of 120 JPY to buy an orange. Then, they selected one of the three oranges to buy. These processes constituted one round. We conducted 12 rounds.

*Step 4*: Each participant filled in the questionnaire after completing all rounds.

*Step 5*: Each participant received a fixed reward. In the online, the research company paid 1 point for 1 question which could be used for buying the goods on the EC site. The process lasted for approximately 15 minutes.

**2.5.4. HOSCT.** The HOSCT treatment had the same steps as the HOS treatment except for Step 1. In Step 1, each participant read an online instruction as well as cheap-talk, presented on different screens. We fixed the time of the cheap-talk screen to 5 seconds to prevent participants from skipping the explanation. The process lasted for approximately 15 minutes.

## 3 Model

### 3.1 The random parameter logit (RPL) model

The RPL model [61,62] is a popular estimation model employed in CE literature. The RPL model relaxes the assumption of the independence of irrelevant alternatives and assumes heterogeneous preferences across participants. This model enhances the accuracy and reliability of the estimated results. In addition, it is based on random utility theory, which is central to CEs. The basic assumption underlying random utility theory is that decision makers maximize utility; thus, the theory assumes that decision makers would select the alternative that maximizes their utility. Although the utility of an alternative for an individual ($U$) cannot be observed, it can be assumed to consist of a deterministic (observable) component ($V$) and a random error (unobservable) component ($\varepsilon$). Formally, an individual $n$'s utility for alternative $i$ in each of the $t$ choice sets can be expressed as $U_{int} = V_{int} + \varepsilon_{int} = \beta'_n X_{int} + \varepsilon_{int}$. The density of $\beta'_n$ is denoted by $f(\beta|\theta)$, where $\theta$ is a vector of the true parameters of the taste distribution. $X_{int}$ denotes the explanatory variables of $V_{int}$ for alternative $i$, individual $n$, and choice set $t$. The random error component $\varepsilon_{int}$ is assumed to follow a Type I extreme value distribution and be independently and identically distributed.

The conditional probability of alternative $i$ for individual $n$ in choice set $t$ is expressed as follows:

$$P_{int}(\beta'_n) = \frac{exp(\beta'_n X_{int})}{\sum_{j=1}^{J} exp(\beta'_n X_{jnt})}. \tag{1}$$

The probability of the observed sequence of choices conditional on knowing $\beta'_n$ is expressed as follows:

$$S_n(\beta'_n) = \prod_{t=1}^{T} P_{i(n,t)nt}(\beta'_n), \tag{2}$$

where $i(n,t)$ represents the alternative selected by individual $n$ from choice set $t$. The unconditional probability of the observed sequence of choices for individual $n$ is the integral of the conditional probability over all possible variables of $\beta'$, and can be expressed as follows:

$$P_n(\theta) = \int S_n(\beta) f(\beta|\theta) d\beta. \tag{3}$$

In most applications, the density $f(\beta|\theta)$ is specified to be normal or log-normal—namely, $\beta \sim N(b,W)$ or $\ln \beta \sim N(b,W)$—where the mean, $b$, and covariance, $W$, are estimated. In this study, we use normal density.

Hence, the main effect in Model 1 and the main effect with the interaction term in Model 2 are estimated using the RPL model with socioeconomic characteristics. Thus, the two indirect utility functions are expressed as follows:

$$\text{Model 1}: \ V_{int} = \beta_{1n}Price_{int} + \beta_{2n}Carbon_{int}. \tag{4}$$

$$\text{Model 2}: \ V_{int} = \beta_{1n}Price_{int} + \beta_{2n}Carbon_{int} + \beta_{3na}Attribute_{int} \times Qestion_{na} \tag{5}$$

where $Price_{int}$ and $Carbon_{int}$ are the price and carbon emissions, respectively, of alternative $i$, individual $n$, and choice set $t$. $Attribute_{int}$ consists of the price and carbon emission variables. $Qestion_{na}$ consists of demographics, the ECCB scale, the frequency of eating oranges, buying goods with an eco-label, and daily environmental behavior. $Attribute_{int} \times Qestion_{na}$ is the interaction term; including it in the model further refines the estimation on individual-specific characteristics regarding the heterogeneity in the mean of the random parameter [2]. The interaction term shows the effect of demographics on indirect utility by interacting with the attributes and enhances the accuracy and reliability of estimates of $V_{ni}$. $\beta_{1n}$, $\beta_{2n}$ and $\beta_{3na}$ are parameters estimated by the respective explanatory variables of the above attributes.

Generally, the RPL model consists of two types of variables—namely, fixed and random variables, which are assumed to be homogenous and heterogeneous preferences, respectively. In our model, $Price_{int}$ and $Carbon_{int}$ are the random variables.

In the RPL models shown in Table 1, prices are set as fixed parameters [6,12,14–16]. However, a fixed cost coefficient is criticized because the assumption that all respondents have the same marginal price utility is unrealistic. Therefore, we used the random parameter model for the price. However, it is still an open question which distribution should be set as the random parameters in the RPL model [63]. In Table 1, only one study used the preference space with normal distribution for the price [11].

However, it has been shown that the moments of the WTP distribution become undefined when a random monetary attribute such as price is assumed to be normally distributed [64]. To avoid this, it is desirable to set the distribution of the random monetary attribute to the log-normal distribution, or to use the WTP space instead of preference space [63]. In Table 1, two studies used the WTP space with negative log-normal distribution for the price [17,18].

There is only one study of Carlson et al. which employed the preference space with normal distribution in the HB studies [11]. To show the additional evidence for the normal distribution in this field, this study employed the preference space with the normal distribution following the basic idea suggested by Train [62].

## 3.2 The marginal WTP

We calculated the marginal WTP for lower carbon emissions using the coefficients in the main effect. Since our model does not include the opt-out option in the choice sets, the calculated WTP is defined as the marginal WTP, rather than the total WTP, which indicates the total benefit. The marginal WTP shows the welfare measures represented by the marginal rate of substitution between the coefficients of the attributes and price. Thus, the calculated WTP is defined as:

$$WTP = -\frac{\beta_j}{\alpha} \tag{6}$$

where $\alpha$ is the coefficient of the estimated $Price$, and $\beta$ is the coefficient of the estimated attributes $j$ except for $Price$. Standard deviations, standard errors, and the 95% confidence-interval bounds are derived using the Krinsky and Robb method. Since the estimated $Price$ is

normalized by the marginal WTP, we directly compare the estimation results of *Carbon* in each treatment.

### 3.3 Hypotheses

According to the complete combinatorial (CC) test [65], we set the null hypothesis of no difference in the estimated mean WTPs between two treatments as follows. This type of comparison has been applied by several studies [5,6,11,14,17,66].

$H_0^1$: $WTP_{NHLEE}$ — $WTP_{HLS}$ = 0, and $H_1^1$: $WTP_{NHLEE}$ — $WTP_{HLS}$ < 0

This hypothesis tests whether the monetary incentive influences HB in the laboratory setting. If $H_0^1$ is rejected, it implies that providing a monetary incentive induces HB in the laboratory setting.

$H_0^2$: $WTP_{HOSCT}$ — $WTP_{HOS}$ = 0, and $H_1^2$: $WTP_{HOSCT}$ — $WTP_{HOS}$ < 0

This hypothesis tests whether cheap-talk influences HB in the online setting. If $H_0^2$ is rejected, it implies that using cheap-talk induces HB in an online survey.

$H_0^3$: $WTP_{HLS}$ — $WTP_{HOS}$ = 0, and $H_1^3$: $WTP_{HLS}$ — $WTP_{HOS}$ < 0

This hypothesis tests whether laboratory settings induce lower WTP than online settings in the hypothetical condition. If $H_0^3$ is rejected, it implies that the laboratory setting reduces WTP, like HB, even in the hypothetical condition. It implies that the former setting makes participants behave more realistically than the latter setting.

$H_0^4$: $WTP_{NHLEE}$ — $WTP_{HOS}$ = 0, and $H_1^4$: $WTP_{NHLEE}$ — $WTP_{HOS}$ < 0

This hypothesis confirms that the non-hypothetical laboratory setting induces HB, compared to the hypothetical online survey setting. If $H_0^4$ is rejected, it implies that experimental economics with a small sample is more realistic than an online survey with a large sample. This is among the main contributions of this study.

$H_0^5$: $WTP_{HOSCT}$ — $WTP_{HLS}$ = 0, and $H_1^5$: $WTP_{HOSCT}$ — $WTP_{HLS}$ < 0

This hypothesis tests whether online settings with cheap-talk induce HB compared to laboratory settings without cheap-talk in the hypothetical condition. If $H_0^5$ is rejected, it implies that an online survey with a large sample is more realistic than a laboratory setting without cheap-talk, even though participants are physically present in a university laboratory for the experiment. This is one of this study's main contributions.

$H_0^6$: $WTP_{NHLEE}$ — $WTP_{HOSCT}$ = 0, and $H_1^6$: $WTP_{NHLEE}$ — $WTP_{HOSCT}$ < 0

This hypothesis checks whether the non-hypothetical laboratory setting without cheap-talk induces HB compared to the hypothetical online survey setting with cheap-talk. If $H_0^6$ is rejected, it implies that a laboratory setting with a small sample is more realistic than an online survey setting with a large sample, even though the survey involves cheap-talk. This is also one of this study's main contributions.

## 4 Results

### 4.1 Samples

In the NHLEE and HLS treatments, we recruited both students from Osaka University and residents living in the area surrounding the university from a variety of demographic backgrounds. To recruit students, we distributed leaflets across campus. To recruit residents, we distributed flyers attached to the four most popular Japanese newspapers to 15,700 households. We conducted 15 sessions with 104 participants (41 students and 63 residents) in the NHLEE. The participants earned an average of 1,407 JPY. Only two men did not take the oranges home. In the HLS, we conducted 19 sessions with 212 participants (96 students and 116 residents).

In the HOS and HOSCT treatments, we controlled the samples in each treatment so that they were representative of the corresponding population with respect to age and gender in Japan. In addition, we assumed that participants were older than 18 years to ensure the same age composition as in the NHLEE and HLS treatments. Rakuten Insight Global distributed the survey to 15,379 people in their panels and recruited 500 people for each treatment.

Table 3 summarizes the demographics of each treatment. In terms of gender, the percentage of women in the lab was higher than online. As for age, most of the lab participants were under 25 years old, while most of the online participants were over 40 years old. Regarding family structure, most of those for lab participants had four or more members, but most online had two or three. For education, college graduates were the most common for both. In terms of income, most Lab participants had less than 2.5 million or more than 7 million, while most Online had more than 7 million, followed by 2.5–4 million. In terms of frequency, lab participants were more likely to eat oranges in a week. The use of public transportation was more than twice as high in lab participants than online participants (50% in labs and 20% in online); Not tap running is higher for lab participants (more than 80% in labs but around 70% in online); Air conditioning is higher for lab participants (more than 70% in labs but around 50% in online); Walking and Garbage was higher in the lab at over 75% and online at about 70%. Overall, lab participants tended to have more heightened environmental awareness and caring behavior than online participants.

Although there are differences in each treatment, as a whole sample, the ratio of men to women was 47% and 53%, with slightly more women than men. The age ratio was 44% for those under 40 years old and 53% for those over 40 years old, with middle-aged respondents accounting for a slight majority. In terms of family structure, many respondents consisted of two to three members, which was the household structure. In terms of education, college graduates were the most common. In terms of income, the largest number of respondents had an income of over $7million, indicating that people with high annual incomes participated in the survey.

## 4.2 The main effect and the WTP: Testing for HB

We employed the panel RPL estimation model using LIMDEP 11.0 and NLOGIT 6.0. First, we tested the hypothesis of equal utility parameters among treatments using the likelihood ratio (LR) test [67]. The LR test employed the log likelihood values in Model 1, estimated using Halton draws with 50 replications, following Carlsson et al. [8], under the grid search procedure [67]. The LR test rejected the hypothesis at the 1% level (LR = 2(-12760.8- (-1020.2–2110.4–4827.0–4760.4)) = 85.6). Thus, we estimated each treatment.

Next, we analyzed the main effects of *Price* and *Carbon* in Model 1, as shown in Table 5. This model was estimated using Halton draws with 500 replications [68,69] to improve the validity. The variables *Price* and *Carbon* are random parameters in the model and specified to be normally distributed [61,68]. We considered that preference for price in this study is not as clear as in other studies [5]. This is because the higher price can be a signal of better quality. The variables *Price* and *Carbon* indicate significantly negative signs at the 1% level in all treatments. These results imply that people prefer to purchase cheaper oranges and oranges with lower carbon emissions, regardless of the setting. The standard deviations of these variables were significant in all treatments, implying that individual preferences for price and carbon footprint were heterogeneous in all treatments.

Finally, we calculated the marginal WTP in each treatment using the Krinsky and Robb method. The marginal WTP per 1g of carbon emission reduction was 0.53 JPY, 0.52 JPY, 0.54 JPY, and 0.58 JPY in the NHLEE, the HLS, the HOS, and the HOSCT, respectively. Additionally, we figured the kernel density distributions of mean marginal WTP for carbon emissions

**Table 5. The random parameter logit regression results in the main effect (Model 1).**

| | Non-hypothetical Lab Economic Experiment (NHLEE), | Hypothetical Lab Survey (HLS) | Hypothetical Online Survey (HOS) | Hypothetical Online Survey with Cheap-talk (HOSCT) |
|---|---|---|---|---|
| *Random parameter* | | | | |
| Carbon | -0.08 *** | -0.08 *** | -0.10 *** | -0.10 *** |
| | (0.01) | (0.01) | (0.01) | (0.01) |
| Price | -0.15 *** | -0.16 *** | -0.18 *** | -0.18 *** |
| | (0.02) | (0.01) | (0.01) | (0.01) |
| *Standard deviation* | | | | |
| Carbon | 0.07 *** | 0.07 *** | 0.10 *** | 0.11 *** |
| | (0.01) | (0.01) | (0.01) | (0.01) |
| Price | 0.12 *** | 0.13 *** | 0.19 *** | 0.20 *** |
| | (0.02) | (0.01) | (0.01) | (0.01) |
| *Marginal WTP* | | | | |
| Carbon: Mean | -0.53 [-0.7, -0.38] | -0.52 [-0.63, -0.43] | -0.54 [-0.6, -0.47] | -0.58 [-0.65, -0.5] |
| Carbon: Standard deviation | 0.08 | 0.05 | 0.03 | 0.04 |
| Log Likelihood | -1020.29 | -2110.42 | -4827.05 | -4760.48 |
| McFadden R2 | 0.25 | 0.24 | 0.26 | 0.27 |
| Observation | 1248 | 2554 | 6000 | 6000 |
| # of respondents | 104 | 212 | 500 | 500 |

Notes: Standard errors are in parentheses. Marginal WTP represents mean values from Krinsky and Robb simulations. Numbers in the blankets show 95% confidential intervals.

***, **, and * denote that the parameters are different from zero at the 1%, 5%, 10% significance levels, respectively. Observations are equal to the # of respondents multiplied by 12 choice sets.

in each treatment, as shown in Fig 4. To test the hypotheses on HB mentioned in Section 3.3, we employed the CC test. The CC test did not reject all hypotheses at the 5% significant levels; $H_0^1$ (p = 0.52), $H_0^2$ (p = 0.76), $H_0^3$ (p = 0.41), $H_0^4$ (p = 0.47), $H_0^5$ (p = 0.78), $H_0^6$ (p = 0.3). Thus, the results do not indicate the presence of HB.

## 4.3 The main effect with the interaction term in investigating carbon footprint

We analyzed the main effect with the interaction term in Model 2 as shown in Table 6. First, we describe the independent variables except for the attributes in Model 2 below. The interaction term with attributes was employed for demographics, the ECCB scale, the frequency of eating oranges, buying goods with an eco-label, and daily environmental behavior, are detailed in Tables 3 and 4. The demographic dummy variables were Female and University. The ordered categorical variables were Age, Household, and Income. Regarding the ECCB scale, the order was changed in the analysis because the higher order was interpreted as agreeing more before estimation. We then assumed *HighECCB* as a dummy variable, assigning it the value of 1 if it was more than the average total score in each treatment and 0 otherwise. *Frequency* was the number of oranges the participants ate per week. *Label* represented participants who bought goods with an eco-label when they purchased goods and was a dummy variable. Regarding daily environmental behavior, six variables, *Not using shopping bags*, *Using public transportation*, *Not running*, *AirConditioning*, *Walking*, and *Separating garbage*, were used as dummy variables.

Next, we explain only the significant results for the interaction terms. The estimation results for all variables in Model 2 are in the Supplementary Materials. For *Carbon×Female*, the

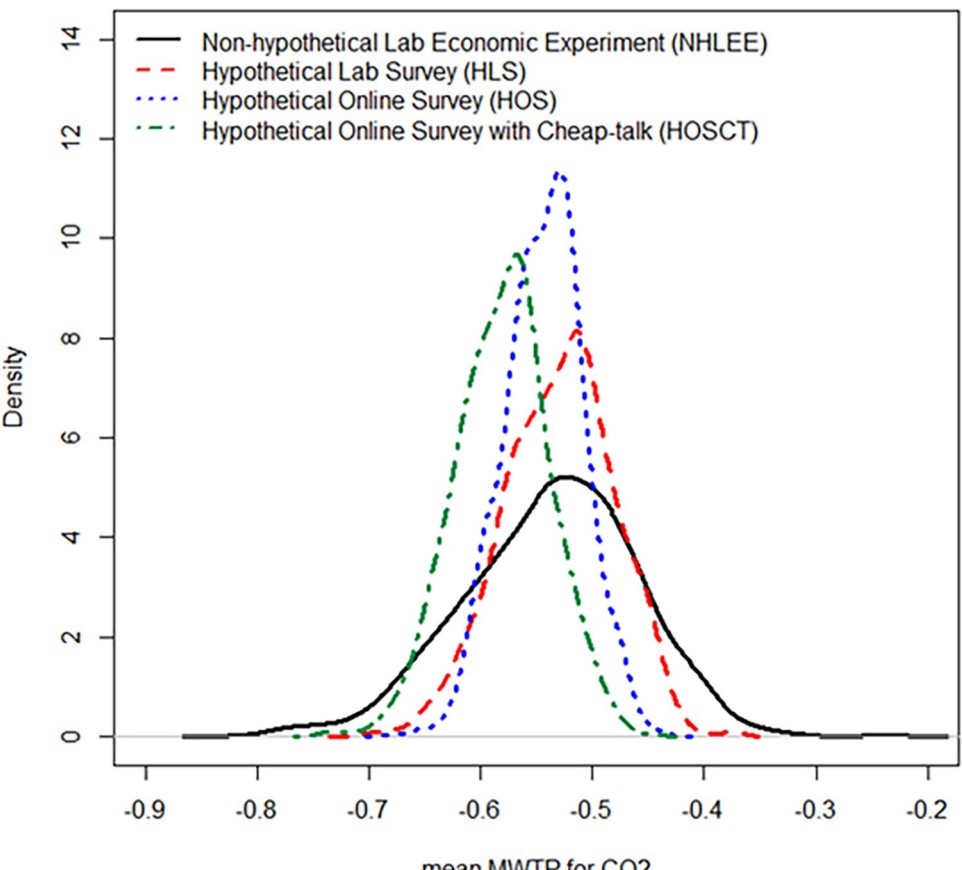

**Fig 4. Mean marginal WTP for carbon emissions in each treatment.**

HOSCT had a significantly negative sign, implying that females prefer oranges with low carbon emissions in the hypothetical condition with cheap-talk. For *Carbon×Age*, the NHLEE and the HLS had significantly positive signs, implying that students prefer oranges with low carbon emissions under the controlled environment monitored by the experimenter. For *Carbon×HighECCB*, all treatments had significantly negative signs, implying that people with high environmental consciousness prefer oranges with lower carbon emissions, regardless of the setting. For *Carbon×Frequency*, the HOSCT had a significantly positive sign, implying that people who sometimes eat oranges prefer oranges with low carbon emissions in the hypothetical condition with cheap-talk. For *Carbon×Label*, the HOS had a significantly negative sign, implying that people who buy goods with an eco-label prefer oranges with low carbon emissions in the hypothetical condition. For *Carbon×Not using shopping bags*, the HOSCT had a significantly negative sign, which implies that people who do not use shopping bags prefer oranges with low carbon emissions in the hypothetical condition with cheap-talk. For *Carbon×Public Transportation*, the HOSCT had a significantly negative sign, which implies that people who often use public transportation prefer oranges with low carbon emissions in the hypothetical condition with cheap-talk. For *Carbon×Garbage*, the HLS and the HOS had significantly negative signs, which implies that people who frequently separate their garbage prefer oranges with low carbon emissions in the hypothetical condition.

For *Price×Female*, the NHLEE, HOS, and HOSCT had significantly positive signs, implying that males prefer a cheaply priced orange in the non-hypothetical laboratory and hypothetical

**Table 6. The RPL estimation results in the main effect with interaction (Model 2) (Only significant variables).**

| | Non-hypothetical Lab Economic Experiment (NHLEE) | Hypothetical Lab Survey (HLS) | Hypothetical Online Survey (HOS) | Hypothetical Online Survey with Cheap-talk (HOSCT) |
|---|---|---|---|---|
| *Random parameter* | | | | |
| Carbon | -0.09 | -0.02 | 0.02 | -0.06 |
| | (0.09) | (0.05) | (0.04) | (0.04) |
| Price | -0.20 * | -0.21 *** | -0.18 *** | -0.25 *** |
| | (0.10) | (0.07) | (0.07) | (0.07) |
| *Heterogeneity in mean* | | | | |
| Carbon × Female | -0.05 | -0.02 | -0.01 | -0.03 ** |
| | (0.03) | (0.02) | (0.01) | (0.01) |
| Carbon × Age | 0.02 * | 0.01 *** | 0.00 | 0.00 |
| | (0.01) | (0.00) | (0.00) | (0.00) |
| Carbon × High ECCB | -0.04 * | -0.03 ** | -0.03 *** | -0.04 *** |
| | (0.02) | (0.02) | (0.01) | (0.01) |
| Carbon × Frequency | 0.01 | -0.01 | 0.00 | 0.02 ** |
| | (0.02) | (0.01) | (0.01) | (0.01) |
| Carbon × Label | 0.01 | 0.00 | -0.04 * | -0.01 |
| | (0.07) | (0.04) | (0.02) | (0.03) |
| Carbon × Not using shopping bags | 0.00 | -0.02 | -0.01 | -0.02 * |
| | (0.02) | (0.02) | (0.01) | (0.01) |
| Carbon × Public Transportation | -0.02 | -0.02 | -0.02 | -0.06 *** |
| | (0.02) | (0.02) | (0.01) | (0.02) |
| Carbon × Garbage | 0.02 | -0.03 * | -0.04 *** | -0.01 |
| | (0.03) | (0.02) | (0.01) | (0.01) |
| Price × Female | 0.06 * | 0.03 | 0.05 ** | 0.04 ** |
| | (0.03) | (0.03) | (0.02) | (0.02) |
| Price × Age | 0.03 *** | 0.01 | 0.02 *** | 0.03 *** |
| | (0.01) | (0.01) | (0.01) | (0.01) |
| Price × University | 0.01 | -0.01 | -0.01 | -0.03 * |
| | (0.03) | (0.02) | (0.02) | (0.02) |
| Price × Income | 0.01 | -0.01 | 0.01 * | 0.01 |
| | (0.01) | (0.01) | (0.01) | (0.01) |
| Price × High ECCB | -0.01 | 0.01 | 0.06 *** | 0.04 ** |
| | (0.03) | (0.02) | (0.02) | (0.02) |
| Price × Label | -0.02 | 0.02 | -0.11 *** | -0.01 |
| | (0.08) | (0.06) | (0.04) | (0.05) |
| Price × Not using shopping bags | -0.01 | -0.06 ** | -0.03 * | -0.07 *** |
| | (0.03) | (0.02) | (0.02) | (0.02) |
| Price × Public Transportation | -0.04 * | 0.02 | -0.02 | -0.06 ** |
| | (0.02) | (0.02) | (0.02) | (0.03) |
| Price × Not tap running | 0.00 | 0.03 | -0.07 *** | -0.01 |
| | (0.05) | (0.03) | (0.02) | (0.02) |
| Price × Walking | 0.00 | 0.02 | 0.00 | 0.04 * |
| | (0.03) | (0.02) | (0.02) | (0.03) |
| *Standard deviation* | | | | |
| Carbon | 0.06 *** | 0.06 *** | 0.09 *** | 0.10 *** |

(*Continued*)

**Table 6.** (Continued)

| | Non-hypothetical Lab Economic Experiment (NHLEE) | Hypothetical Lab Survey (HLS) | Hypothetical Online Survey (HOS) | Hypothetical Online Survey with Cheap-talk (HOSCT) |
|---|---|---|---|---|
| | (0.01) | (0.01) | (0.01) | (0.01) |
| Price | 0.06 *** | 0.12 *** | 0.18 *** | 0.18 *** |
| | (0.01) | (0.01) | (0.01) | (0.01) |
| Log Likelihood | -852.69 | -1820.09 | -4776.35 | -4713.43 |
| McFadden R2 | 0.24 | 0.25 | 0.27 | 0.28 |
| Observation | 1248 | 2554 | 6000 | 6000 |
| # of respondents | 104 | 212 | 500 | 500 |

Notes: Standard errors are in parentheses.

***, **, and * denote that the parameters are different from zero at the 1%, 5%, 10% significance levels, respectively.

online settings. For *Price×Age*, the NHLEE, the HOS, and the HOSCT had significantly positive signs, implying that younger people prefer cheaply priced oranges in the non-hypothetical laboratory and hypothetical online settings. For *Price×University*, the HOSCT had a significantly negative sign, which implies that university graduates prefer cheaply priced oranges in the hypothetical condition with cheap-talk. For *Price×Income*, the HOS had a significantly positive sign, which implies that low-income earners prefer cheaply priced oranges in the hypothetical online setting. For *Price×HighECCB*, the HOS and the HOSCT had significantly positive signs, implying that people with low environmental consciousness prefer cheaply priced oranges in the hypothetical online setting. For *Price×Lables*, the HOS had a significantly negative sign, implying that people who buy goods with an eco-label prefer cheaply priced oranges in the hypothetical online setting. For *Price×Not using shopping bags*, the HLS, the HOS, and the HOSCT had significantly negative signs, which implies that people who use shopping bags prefer cheaply priced oranges in the hypothetical condition. For *Price×Public Transportation*, the NHLEE and the HOSCT had significantly negative signs, which implies that people who use public transportation prefer cheaply priced oranges in the actual condition as the non-hypothetical laboratory and hypothetical online settings. For *Price×Not tap running*, the HOS had a significantly negative sign, which implies that people who frequently do not leave the tap running and turn off the light after exiting a room prefer cheaply priced oranges in the hypothetical online setting. For *Price×Walking*, the HOSCT had a significantly positive sign, which implies that people who walk less in their daily life prefer cheaply priced oranges in the hypothetical condition with cheap-talk. As a result above, only *Carbon×HighECCB* demonstrated the same result in all treatments; we offer our reasoning for the HB in the following section.

## 5. Discussions

This study set the attributes of orange as Price and Carbon emissions and examined HB by comparing four treatments. A WTP analysis results did not show HB in any treatment comparisons. Meanwhile, results for the interaction term evidenced that people with high ECCB scales prefer oranges with low carbon footprints.

### 5.1 Comparison of studies employing common environments for non- and hypothetical conditions

In this study, comparisons between laboratory settings, between online settings, and between laboratory and online settings were examined. We will discuss the effect on HB, focusing on

the research environment. Environments for comparing non- and hypothetical conditions in Table 1 are classified into two types: common environment and different environment. In the common environment, there are two conditions: the laboratory setting and the non-laboratory setting.

**5.1.1 Laboratory setting.** Two previous studies were conducted in the laboratory setting. Lusk and Schroeder compared the non-hypothetical and hypothetical conditions in a university laboratory setting [6]. Their results did not show HB toward the following attributes of beef: Generic, GT, Natural, Choice, or CAB. Meanwhile, De-Magistris et al. compared the non-hypothetical and hypothetical conditions in a town laboratory setting in relation to almonds [14]. Their results showed HB toward the attributes of Organic and Food mileage labeling. However, in the comparison of the hypothetical and hypothetical with cheap-talk conditions, Food mileage no longer showed HB. In this study, the comparison between the laboratory settings ($H_0^1$) showed no HB, supporting Lusk and Schroeder [6]. Additionally, our results for the hypothetical and hypothetical with cheap-talk conditions also supported De-Magistris et al. [14]. However, the results of the non-hypothetical and hypothetical conditions were not supported. Apart from food studies, Carlsson and Martinsson compared HB in a university laboratory [20]. Their results did not show HB toward the attribute of Donation in relation to environment projects of the World Wide Fund for Nature (WWF). Johansson-Stenman and Svedsäter compared HB in college laboratories [22]. Their results did not show HB in the attribute of Campaign against the WWF's donations for the protection of elephants in Asia. Therefore, neither of these studies showed any HB for environmental or climate change attributes. Similar to previous studies on laboratory settings, this study had difficulty observing HB toward environmental and climate change attributes, perhaps because they may have characteristics very different from taste; that is, since the impact on consumer preferences is less likely in the first place, the HB is unlikely to be observed. Future research would do well to address this challenge.

**5.1.2 Non-laboratory setting.** Previous studies in which HB comparisons were conducted in non-laboratory settings are divided into two types: non-face-to-face and face-to-face. To begin, we will describe the non-face-to-face type. Carlsson et al. used postal surveys to compare the hypothetical and hypothetical with cheap-talk conditions [11]. While the GMO attribute for beef showed HB, other attributes, such as Improved labelling, Out all year, and Mobile did not show HB. Liebe et al. conducted non- and hypothetical online choice experiments without cheap-talk by a professional survey organization and showed HB against attributes of Organic and Fair trade in tea [18].

On the other hand, we used online surveys. $H_0^2$ compared the hypothetical and hypothetical with cheap-talk conditions and did not show HB. Because Carlsson et al.'s study [11] as well as this element of our own used non-face-to-face methods the effect of cheap-talk has a chance to be weak in such an environment HB is difficult to find. Liebe et al. suggests that we have a chance to show HB for carbon footprint if we have an online experiment [18]. Next, we will describe the face-to-face type. Moser et al. employed an interview at a supermarket and conducted several treatments to compare the hypothetical and hypothetical with cheap-talk conditions [16]. They report that HB was present in some appearance-related attributes for apples; however, other attributes for apples, such as Method of production, Origin, Reduced impact on climate, did not show HB. They also compared the non- and hypothetical conditions in the same environment and found HB in some levels of Cultivation methods, Appearance, and Origin began to show HB. However, Reduced impact on climate still failed to show HB. Alemu and Olsen conducted an interview with town residents, comparing the hypothetical condition with a cheap-talk condition and the non- hypothetical condition [17]. In both comparisons, the attributes, Amount of cricket flour and Whether some portion of the wheat

flour is fortified or not, showed HB. These face-to-face studies suggest that HB is hard to find in the attributes of environmental and climate change, as in laboratory settings. However, the effect of cheap-talk scripts cannot be generalized.

## 5.2 Comparison of studies employing different environments for non- and hypothetical conditions

Next, we will explain previous studies studying HB by comparing non- and hypothetical conditions under different environments. In a previous study comparing laboratory setting experiments with face-to-face street surveys, Aoki et al. showed HB for ham sandwiches with the Sodium nitrate attribute [4]. Meanwhile, in a study that compared online store experiments with in-person store surveys, Wuepper et al. found no HB for coffee with the attribute of Water footprint label [19]. However, this HB analysis used marginal effects instead of WTP. $H_0^3$–$H_0^6$ in this study compared laboratory and online settings, but neither showed HB. These comparisons suggest that environmental attributes tend not to show HB. However, to date, few studies have compared laboratories with other settings; thus, further research is necessary to confirm this trend.

## 5.3 Attributes and psychological factors

The previous sections note that environmental and climate change attributes seem unlikely to exhibit HB, regardless of the setting; similarly, in this study, HB was also not observed with environmental attributes. This section discusses our results from the ECCB scale. Our analysis using the interaction term revealed that people with high ECCB prefer a low carbon footprint over people with low ECCB in all treatments; this suggests that environmental awareness of carbon footprints is not affected by treatment type. Therefore, there may have been no HB in relation to the carbon footprint regardless of treatment.

Like us, Grebitus, Lusk, et al. examined HB using psychological measures, verifying HB against the food miles attribute for apples and wine by considering the Big Six personality traits of Openness, Conscientiousness, Extraversion, Agreeableness, Neuroticism, and Agency [15]. They found that, in the hypothetical condition for apples, participants with Openness, Neuroticism, Extraversion, and Conscientiousness showed a lower probability of choosing the opt-out option, which was an option that does not include Food miles. However, in the non-hypothetical condition for apples, only Agency had a similar effect. In the results for wine, participants with Neuroticism, Extraversion, and Conscientiousness in the hypothetical condition showed a lower probability of choosing the opt-out option, and no psychological measure showed a similar effect in the non-hypothetical conditions. Ultimately, the psychological impact on food preferences attributable to food miles was stronger in the hypothetical condition than in the non-hypothetical condition, which suggests that food miles affected the HB—notably, this result differs from our own; the reasons for this asymmetry remain unclear.

Scholars believe that HB emerges for two reasons: differences in treatments influence participant psychology and psychologies of participants in each treatment differ in the first place. Identifying which reason causes HB in a particular case is a challenge for future research. In future studies examining HB, prior filtering of participants may be important in between-subject designs with regard to psychology.

Regarding other factors, the common interactions were Carbon × Age in the experiment, and Price × Female, Price × Age, Price × High ECCB, and Price × Not using shopping bags in the online sample. Compared to the experiment, the online sample has a more adjusted demographic ratio, so it is possible that the experiment could produce the same results if the same

demographic ratio was used in the same way as the online survey. Other than these, the fact that the interactions differed between treatments is an issue for future work.

The reason "not leaving the tap running" and "walking", which were significant in PRICE, were not significant in CFT is that these two behaviors strongly reflect the awareness of saving money rather than $CO_2$ reduction behavior or good behavior for the environment. On the other hand, the reason why "separating garbage," which was significant in the CFT, was not significant in the PRICE is thought to be that garbage separation is significant because it affects the increase or decrease of $CO_2$ from the viewpoint of waste incineration, but garbage separation is not significant in the PRICE because it does not lead to saving behavior.

Additionally, low-income earners prefer cheaply priced oranges only in HOS which is the treatment most prone to virtual bias compared to the other treatments. Only one previous study, Van Loo et al. found the interaction with price and high income positive and significant in a hypothetical survey of cheap talk [40]. Since income generally has a strong influence on purchasing power, one possibility is that income may have a strong influence on the price in situations where no constraints are placed on it by the experimenter and nothing is said about it. However, the question of how treatment affects attributes is still open because, as mentioned earlier, there are no other studies comparing the two.

## 5.4 Limitations

There are three limitations. The first is the sample selection bias between treatments. To compare the laboratory treatments (NHLEE, HLS) and the online treatments (HOL, HOLCT), the responses had to be collected in different ways. In experiments, it is very difficult to recruit hundreds of subjects according to demographic ratios. Therefore, there is a difference in the demographic ratio and sample size between the online survey and the laboratory experiment and survey. For this reason, we tried to reduce the bias as much as possible by examining the interaction effect with personal attributes in the analysis. Future tests of the difference in HB should ensure that there is no difference in the subject samples between treatments.

Secondly, as a treatment, we did not do a laboratory survey with CT and non-hypothetical online experiment. If we do these two, we can cover all six treatments: implementation environment × CT presence/absence × monetary incentive presence/absence = 6 treatments. It is relatively easy to communicate CT in the lab. To do a non-hypothetical online experiment is a challenge, as research companies often pay monetary points instead of rewarding with cash, which is against the induced value theory suggested by Vernon Smith.

Third, in the method, we calculated WTP using preference space; since we needed to compare WTP for HB validation, it would be preferable to estimate WTP directly in WTP-space, considering scale effects between treatments. However, we tried to estimate WTP-space from the results of this study using the generalized mixed logit model of NLOGIT, but the estimation was not possible. An appropriate design for estimation in WTP-space is an issue to be explored in future research.

## 6. Conclusions

This study was conducted in laboratory and online settings, and neither treatment revealed HB in relation to the carbon footprint of oranges. Hence, we conclude that environmental and climate change attributes in food are probably less prone to HB. In addition, an analysis using the environmental psychology scale confirmed the trend, reducing the coefficients of carbon emissions in all treatments because the consumer's environmental awareness scale was not affected by all treatments. Accordingly, when testing for HB going forward, it is important to consider not only treatment differences but also the effects of psychological scales related to

the target attribute. In sum, this study confirmed that policymakers would do well to consider psychology as well as attributes, rather than thinking only about whether to use an economic experiment or a survey.

## Supporting information

**S1 Table. The random parameter logit regression results in the main effect (Model 1) and that with interaction (Model 2) (All).**
(DOCX)

**S1 File. Instraction used in the NHLEE treatment.**
(DOCX)

**S2 File. Cheap-talk script used in the HOSCT treatment.**
(DOCX)

**S1 Dataset.**
(XLSX)

## Acknowledgments

We appreciate the helpful comments of the participants at the EFFoST 2017 conference in Vienna, Austria, and the Eurosence 2018 conference in Verona, Italy. We thank Professor William H. Greene and the LIMDEP/NLOGIT list members for useful comments on program codes in LIMDEP. We thank Professor Kiyokazu Ujiie and the anonymous reviewers for their suggestions to revise the manuscript as well as the following staffs, Kenta Onoshiro, Yoko Terada, and Katsuyuki Aoki.

## Author Contributions

**Conceptualization:** Keiko Aoki, Kenju Akai.

**Data curation:** Keiko Aoki, Kenju Akai.

**Formal analysis:** Keiko Aoki, Kenju Akai.

**Funding acquisition:** Keiko Aoki.

**Investigation:** Keiko Aoki, Kenju Akai.

**Methodology:** Keiko Aoki, Kenju Akai.

**Project administration:** Keiko Aoki, Kenju Akai.

**Visualization:** Keiko Aoki, Kenju Akai.

**Writing – original draft:** Keiko Aoki, Kenju Akai.

**Writing – review & editing:** Keiko Aoki, Kenju Akai.

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
