## [Decision Letter · Decision Letter 0]

15 Jul 2021

PONE-D-21-18475

Testing hypothetical bias in a choice experiment: An application to the value of the carbon footprint of mandarin oranges

PLOS ONE

Dear Dr. Aoki,

Thank you for submitting your manuscript to PLOS ONE. After careful consideration, we feel that it has merit but does not fully meet PLOS ONE’s publication criteria as it currently stands. Therefore, we invite you to submit a revised version of the manuscript that addresses the points raised during the review process.

As you will see you received two reviews very detailed. My own opinion about your actual version of the manuscript is very close to the one expressed by the second reviewer. SO please take particular care in addressing those comments fully.

We look forward to receiving your revised manuscript.

Kind regards,

Luigi Cembalo, PhD

Academic Editor

PLOS ONE

Journal Requirements:

Reviewers' comments:

Reviewer's Responses to Questions

**Comments to the Author**

1. Is the manuscript technically sound, and do the data support the conclusions?

Reviewer #1: Yes

Reviewer #2: Partly

2. Has the statistical analysis been performed appropriately and rigorously? 

Reviewer #1: Yes

Reviewer #2: N/A

3. Have the authors made all data underlying the findings in their manuscript fully available?

Reviewer #1: Yes

Reviewer #2: Yes

4. Is the manuscript presented in an intelligible fashion and written in standard English?

Reviewer #1: Yes

Reviewer #2: Yes

5. Review Comments to the Author

Reviewer #1: The authors investigate the issue of Hypothetical Bias in online Choice Experiments in a case study involving choices among oranges with different environmental impact. I think the paper is of interest both for the methodological aspect and the empirical case study and a valuable contribution to the literature. However, I think it needs to be revised before being suitable for publication, especially as it concerns the econometric analysis of choices. I report my comments in the attachment.

Reviewer #2: This paper investigates the hypothetical bias for the carbon footprint information of mandarin organs in Japan. The hypothetical bias in this paper is defined as the difference in the willingness to pay for product attributes between hypothetical and non-hypothetical conditions in a choice experiment. The authors employed four treatments setting during data recruitment and applied random parameter logit models to analyze the data. Below please find my comments:

• Abstract is cumbersome and lengthy. Please shorten the abstract and only present concise messages.

• Lack of transparencies in each experimental procedure:

a) In lines 243-244 the authors described that each participant received JPY120 per round and there are 12 rounds in total in HOS setting, which is NOT corresponding to the info presented in table 2 (saying that no monetary incentive given in HOS)

b) What are the fixed rewards in HOS and HOSCT settings, respectively?

c) If the HOS and HOSCT are operated via an online survey with recruited panels 500 each. Please describe in detail how did the authors issue the fixed rewards to their online panels. I assume that the online panel were anonymous. However, if the personal information of online panels is requested for receiving rewards, how were the procedure taken ethically? Please describe how it works in detail.

d) Information regarding the average time duration for the experiments for HLS/HOS/HOSCT are missing

• I did not find the matching info used to derive the hypotheses for HB based on the complete combinatorial test published by Poe et al. (2005). Please give scientific sources how did the authors come up with the six hypotheses in pages 20-22.

• Table 1 is not self-explaining: a) please minimize the use of parentheses. b) What are the differences for the info of “(#of subject)” in the 2nd and the 4th columns? Can the 3rd column have labeled as “mode and location of incentive compatible setting”? what is GT and CAB?

• Table 3: a) more description regarding the results of table 3 on page 23 is needed. b) In addition, please provide a column presenting the descriptive results of the aggregated sample, c) what is the category of “others” meant underlying the question of Eco behavior in daily life?

• What do you mean by the term “continuous dummy variables” in line 406?

• How did you overcome the problem of sample selection bias?

• What are the criteria used in the survey for ensuring the eligibility of the participants? E.g. the awareness/recognition of the carbon footprint info?

• Table 6: please explain the reasons why “no tap running” and “walking” did not interact with “Carbon”, and “Garbage” did not interact with “price” in the RPL model? Why the main effects of the used dummy variables were not included?

• if each participant answered 12 choice sets with 3 alternatives each. For example, for a sample size of 104 in NHLEE setting, please explain why did you come up with observations=1248 (table 5 & table 6).

• Line396: I did not find references [7,8] applying CC test for the comparison of differences between the estimated mean WTPs.

• If the authors did not provide an opt-out option in their DCE, it implies that participants are forced to buy orange in each round where the min. purchase price is JPY25 per orange. Given the endowment with JPY120 per round and 12 rounds in total (overall endowment JPY1440) in NHLEE setting, please explain how did you come up with an average of JPY1407 (line364) if participants are rewarded based on the sum of the residuals of the endowment minus the price of the selected orange in each round?

• Line 413: I did not find the dummy variable of “control” included in the RPL model.

• All studies have limitations. What are the limitations in this study? I suggest to focus on the issues regarding method and study design.

• Scientific references are missing in the following statements: lines 3-7 / lines 13-15 / lines 16-17 / lines 66-68 / lines 70-81 / lines 76

• Language issue: line 555

6. PLOS authors have the option to publish the peer review history of their article (what does this mean?). If published, this will include your full peer review and any attached files.

Reviewer #1: No

Reviewer #2: No

---

## [Author Response · Author response to Decision Letter 0]

6 Nov 2021

If the text file below is difficult to read, please see the attached file at the end.

Response to the Review Comments to the Author

Thank you very much for the helpful comments. We have made changes accordingly which we feel have greatly improved the manuscript.

We made changes in the manuscript in red for reviewer #1 and blue for reviewer #2. 

Reviewer #1: The authors investigate the issue of Hypothetical Bias in online Choice Experiments in a case study involving choices among oranges with different environmental impact. I think the paper is of interest both for the methodological aspect and the empirical case study and a valuable contribution to the literature. However, I think it needs to be revised before being suitable for publication, especially as it concerns the econometric analysis of choices. I report my comments in the attachment.

The authors investigate the issue of Hypothetical Bias in online Choice Experiments in a case study involving choices among oranges with different environmental impact. I think the paper is of interest both for the methodological aspect and the empirical case study and a valuable contribution to the literature. However, I think it needs to be revised before being suitable for publication, especially as it concerns the econometric analysis of choices. Below my comments.

1. The transition between the review on carbon footprint studies and the outline of paper structure (lines 107-108) feels rather abrupt. I would add a sentence summarizing the findings of the literature and explaining what the authors expected to find in their study based on previous ones.

Reply: Thank you for pointing this out. Since the preference for CFP is known to be influenced by other eco-labels and environmental awareness, a secondary objective of this study was to examine the demographic and social background factors that promote CFP in the real world. Therefore, we have inserted the following:

p.10-11, l.106-109,

Since these studies found that other eco-labels and environmental awareness affect preference for CFP, this study also analyzed whether personal attributes, environmental psychology, and environmental consideration scales affected preference for CFP to examine the social factors that promote CFP.

2. I may have missed it, but I could not find the indication of the year in which the lab experiments took place. Given that the online surveys were carried out in 2013, I imagine several years could have passed between the surveys and the lab experiments. If that is the case, I think there should be a short discussion in the paper on whether this could have influenced the results.

Reply: Thank you for your suggestion. I have filled in the years of implementation. The lab experiment was conducted in 2012 and the online survey was conducted in 2016. The Consumer Price Index (CPI) for 2012 and 2016 was 97.2 and 99.9, respectively, with 2015 being 100. 2012-2016 was a sluggish period in Japan, as it was below 100, and the situation was similar for the previous eight years. In addition, the inflation rate announced by the IMF was -0.06% in 2012 and -0.12% in 2016, both of which were implemented during a period of decline. Thus, Japan's economic situation has been implemented at a similar time. This was added to the section 2.1 Experimental design. 

p.12, l.136-142,

NHLEE and HLS were conducted in 2012 and HOS and HOSCT were conducted in 2016. During the last eight years, CFP was not popularized, and no laws were made about it. The Consumer Price Index (CPI) [47] for 2012 and 2016 were 97.2 and 99.9, respectively, with 2015 being 100. 2012-2016 was a sluggish period in Japan's prices, as it was below 100, and the situation was similar for the previous eight years. In addition, the inflation rate announced by the IMF [48] was -0.06% in 2012 and -0.12% in 2016, both of which were implemented during a period of decline. Thus, Japan's economic situation has been implemented at a similar time.

3. How were the priors for the D-efficient design retrieved?

Reply: We used the Design Expert software to create a D-optimal design. The software uses the coordinate-exchange algorithm (CEA) to create the design by entering the attributes, their levels, and the type of design. The algorithm calculates the variance-covariance matrix to be minimized. We inserted the following in 2.3 Choice set design. 

p.15, l.184-186,

Design Expert 7.0 (State Easy) was used to design the choice set, with D-optimal as the design type and the coordinate-exchange algorithm (CEA) as the efficient search algorithm. CEA is known to work very well under multinomial logit discrete choice models [58]. 

4. The issues of having a random cost coefficient in RPL models is poorly discussed in lines 298-304. You can refer to Mariel et al. (2021) for a recent discussion on this.

First of all, the cost coefficient "needs" to be negative for results to be consistent with the economic theory, as a positive coefficient would imply that utility increases when cost increases. Secondly, the actual issue with a randomly distributed cost coefficient is the possibility of retrieving positive tails of the distribution (with the above issue) and tails extremely close to zero, which make WTP values explode. At this point of the literature, I would not say that the conventional way to bypass such issues is a fixed cost coefficient, as the authors state. A fixed cost coefficient is criticizable because it implies the assumption that all respondents have the same marginal utility of income (very unrealistic). A much more common solution, nowadays, is to use a lognormal distribution with changed sign (to avoid positive tails), which still has the problem of possible points of the distribution very close to zero. This can be further avoided by adopting specifications in WTP-space (Train and Weeks 2005; Scarpa et al. 2008). 

Given the above and that the main purpose of the study is to compare WTP values across treatments, I would advise to estimate the models in WTP-space rather than in preference space, as models in WTP-space have been consistently found to yield more reliable WTP estimates (again see Mariel et al. (2021), chapter 6, for a recent discussion on this). I also note that nowadays is uncommon to use a normal distribution for cost (as done by authors) for the issues discussed in the previous point, but given that no positive tails were retrieved (based on Figure 4) it should not be an issue in this case. However, if you decide to change to a WTP-space specification, I would advise to use a log-normal distribution for the price/scale parameter.

Reply: Thank you very much for your helpful suggestion. We tried to do the WTP-space estimation, but the program does not converge so the estimation failed. Following your idea, we refer to Mariel et al. (2021) and explain the reason why we choose the normal distribution. According to the literature shown in Table 1, only two studies used WTP space and others employed preference space. Out of studies with preference space, only one study used normal distribution and the others used fixed price. In this manuscript we chose to use the normal distribution because it does not have a positive tail. The results from the fixed price model were added in the supplemental materials. We added the explanation below in 3.1 The random parameter logit (RPL) model:

p.22-23, l. 318-332,

In the RPL models shown in Table 1, prices are set as fixed parameters [6, 12, 14-16]. However, a fixed cost coefficient is criticized because the assumption that all respondents have the same marginal price is unrealistic. Therefore, we used the random parameter model for the price. However, it is still an open question which distribution should be set as the random parameters in the RPL model [60]. In Table 1, only one study used the preference space with normal distribution for the price [11].

However, it has been shown that the moments of the WTP distribution become undefined when a random monetary attribute such as price is assumed to be normally distributed [61]. To avoid this, it is desirable to set the distribution of the random monetary attribute to the log-normal distribution or to use the WTP space instead of preference space [60]. In Table 1, two studies used the WTP space with negative log-normal distribution for the price [17, 18]. 

Tthere is only one study of Carlson et al. which employed the preference space with normal distributioon in the HB studies [11]. To improve the the knowledge of normal distribution in this field, this study employed the preference space with the normal distribution following the basic idea suggested by Train (2005) [59]. 

5. I find interesting that most of the covariates used to explain heterogeneity via interactions have significant effect only in some treatments, i.e. the effect is inconsistent across treatments. I think this deserves some discussion in the paper. 

Reply: Thank you for pointing that out. We agree with your opinion. The results of the interactions explained the results in 4.3 only for variables that are significant for all treatments. However, since there is no previous study in Table 1 that analyzed the interactions with demographic for each treatment, the effect of treatment differences on the interactions with demographics are still an open question. We added the following to the discussion.

p.38-39, l.612-617,

Regarding other factors, the common interactions were Carbon × Age in the experiment, and Price × Female, Price × Age, Price × High ECCB, and Price × Not using shopping bags in the online sample. Compared to the experiment, the online sample has a more adjusted demographic ratio, so it is possible that the experiment could produce the same results if the same demographic ratio was used in the same way as the online survey. Other than these, the fact that the interactions differed between treatments is an issue for future work.

6. Also about the covariates, I find surprising that low-income earners prefer cheaply priced oranges only in the hypothetical online setting, where no real money is involved. How would the authors explain this?

Reply: The part you mentioned is a very interesting result. HOS without CT is the treatment most prone to virtual bias compared to the other treatments. The only previous study, Van Loo et al., found Price*HighIncome to be positive and significant in a hypothetical survey with cheap talk. Since income generally has a strong influence on purchasing power, one possibility is that income may have a strong influence on price in situations where no constraints are placed on it by the experimenter and nothing is said about it. However, how treatment affects attributes is a still open question because, as I mentioned earlier, there are no other studies comparing the two. We have added the following to the discussion.

p.39-40, l.625-632,

Additionally, low-income earners prefer cheaply priced oranges only in HOS which is the treatment most prone to virtual bias compared to the other treatments. Only one previous study, Van Loo et al. (2014) found the interaction with price and high income to be positive and significant in a hypothetical survey of cheap- talk. Since income generally has a strong influence on purchasing power, one possibility is that income may have a strong influence on price in situations where no constraints are placed on it by the experimenter and nothing is said about it. However, how treatment affects attributes is a still open question because, as mentioned earlier, there are no other studies comparing the two.

7. The writing could use some improvements, there are several poorly constructed sentences throughout the text.

Reply: The manuscript has been reviewed by a professional editing company to correct and polish the language

Reviewer #2: This paper investigates the hypothetical bias for the carbon footprint information of mandarin organs in Japan. The hypothetical bias in this paper is defined as the difference in the willingness to pay for product attributes between hypothetical and non-hypothetical conditions in a choice experiment. The authors employed four treatments setting during data recruitment and applied random parameter logit models to analyze the data. Below please find my comments:

1. Abstract is cumbersome and lengthy. Please shorten the abstract and only present concise messages.

Reply: We modified and shortened the Abstract.

2. Lack of transparencies in each experimental procedure: a) In lines 243-244 the authors described that each participant received JPY120 per round and there are 12 rounds in total in HOS setting, which is NOT corresponding to the info presented in table 2 (saying that no monetary incentive given in HOS)

Reply: a) Thank you for pointing this out. HOS is a hypothetical situation, so the choice is based on the assumption that you have 120 yen, i.e., the participants just imagine the scenario. We added the explanation below:

 p.19, l.257,

Each participant imagined that they had an endowment of 120 JPY. Then, they selected one of the three oranges to buy.

b) What are the fixed rewards in HOS and HOSCT settings, respectively?

Reply: HOS and HOSCT are paid by the survey companies at the rate of one point per question. That point can be converted to a point that can be used on the EC site. We added this explanation:

p.19, l.261-263,

In the online, the research company paid 1 point for 1 question which could be used for buying the goods on the EC site. The process lasted for approximately 15 minutes.

c) If the HOS and HOSCT are operated via an online survey with recruited panels 500 each. Please describe in detail how did the authors issue the fixed rewards to their online panels. I assume that the online panel were anonymous. However, if the personal information of online panels is requested for receiving rewards, how were the procedure taken ethically? Please describe how it works in detail.

Reply: Payment was made by the survey company. For this reason, we do not have access to the personal information of each respondent. We followed the national privacy policy and there are no ethical issues. We added this explanation

p.12, l.128-133,

For the online survey, the company recruited the respondents, and managed the participant identities. The authors did not know any personal identification for any respondents. The data collection complied with the Law on the Protection of Personal Information in Japan. The institutions and universities to which the authors belong do not require ethical approval for science research, except in instances that could be deemed life-threatening or harmful to human subjects.

d) Information regarding the average time duration for the experiments for HLS/HOS/HOSCT are missing

Reply: Thank you, this is very important information. We added it.

p.18, l.250, p.19, l.262-263, and l.269

In the HLS, the process lasted about 45 minutes. In the HOS/HOSCT, it lasted about 15 minutes.

3. I did not find the matching info used to derive the hypotheses for HB based on the complete combinatorial test published by Poe et al. (2005). Please give scientific sources how did the authors come up with the six hypotheses in pages 20-22.

Reply: Thank you for pointing this out. Each hypothesis stated on pages 20-22 is the null hypothesis for the statistics. All six possible combinations of choosing 2 out of the 4 treatments were considered. From previous studies, we believe that there is no HB for environmental attributes when DCE is used, but previous studies do not cover all combinations of treatments in this study, so we cannot formulate a theoretical hypothesis for this study. Therefore, the research stance of this study was to investigate if there is a difference rather than to conduct a test to verify the hypothesis. Therefore, the six null hypotheses only formally indicate that there is no difference between the two hypotheses. We modified the explanation below:

p.24, l.349-351,

According to the complete combinatorial (CC) test [65], we set the null hypothesis of no difference in the estimated mean WTPs between two treatments as follows. This type of comparison has been applied by several studies [5, 6, 11, 14, 17, 66].

4. Table 1 is not self-explaining: a) please minimize the use of parentheses. b) What are the differences for the info of “(#of subject)” in the 2nd and the 4th columns? Can the 3rd column have labeled as “mode and location of incentive compatible setting”? what is GT and CAB?

Reply: a) We modified these sentences and eliminated the parentheses in the second and third rows. 

b) There is a difference in sample size between non- and hypothetical treatments. Surprisingly, there is not an equal sample size between the two treatments. We followed your idea and made modifications. GT is “Guaranteed Tender” and CAB is “Certified Angus Beef”. The relevant part of the table has been corrected.

5. Table 3: a) more description regarding the results of table 3 on page 23 is needed. b) In addition, please provide a column presenting the descriptive results of the aggregated sample, c) what is the category of “others” meant underlying the question of Eco behavior in daily life?

Reply: a) We followed your idea and added the explanation below:

p.27, l.401-414,

In terms of gender, the percentage of women in the lab was higher than online. As for age, most of the lab participants were under 25 years old, while most of the online participants were over 40 years old. Regarding family structure, most of those for lab participants had four or more members, but most online had two or three. For education, college graduates were the most common for both. In terms of income, most Lab participants had less than 2.5 million or more than 7 million, while most Online had more than 7 million, followed by 2.5-4 million. In terms of frequency, lab participants were more likely to eat oranges in a week. The use of public transportation was more than twice as high in lab participants than online participants (50% in labs and 20% in online); Not tap running is higher for lab participants (more than 80% in labs but around 70% in online); Air conditioning is higher for lab participants (more than 70% in labs but around 50% in online); Walking and Garbage was higher in the lab at over 75% and online at about 70%. Overall, lab participants tended to have more heightened environmental awareness and caring behavior than online participants.) We added all the samples in Table 3 and added the explanation below:

b) We followed your idea and added the explanation below:

p.28, l. 415-421,

Although there are differences in each treatment, as a whole sample, the ratio of men to women was 47% and 53%, with slightly more women than men. The age ratio was 44% for those under 40 years old and 53% for those over 40 years old, with middle-aged respondents accounting for a slight majority. In terms of family structure, many respondents consisted of two to three members, which was the household structure. In terms of education, college graduates were the most common. In terms of income, the largest number of respondents had an income of over $7million, indicating that people with high annual incomes participated in the survey.

c) In addition to the six Ecobehavior questions, we allowed the participants to freely describe any other activities they were doing. The percentage of those who filled in this questionnaire is shown. The markings have been changed to make it easier to understand. But that variable is not included in the analysis.

In Table 3,

Other eco behaviors apart from the above six questions (free-writing)

6. What do you mean by the term “continuous dummy variables” in line 406?

Reply: Thank you for pointing this out. It is a mistake. Corrected, it is ordinal category variables. We added the question and answers in Table 3.

p.30, l.454-455,

The ordered categorical variables are Age, Household, and Income.

7. How did you overcome the problem of sample selection bias?

Reply: We were not able to overcome sample selection bias. For this reason, we used personal attributes in our analysis. We also included this point in the section on limitations.

8. What are the criteria used in the survey for ensuring the eligibility of the participants? E.g. the awareness/recognition of the carbon footprint info

Reply: We did not use any criteria. Any participants more than 20 years old could join this survey.

9. Table 6: please explain the reasons why “no tap running” and “walking” did not interact with “Carbon”, and “Garbage” did not interact with “price” in the RPL model? Why the main effects of the used dummy variables were not included?

Reply: Thank you for pointing this out. The reason why "not tap running" and "walking", which were significant in PRICE, were not significant in CFT is because these two behaviors strongly reflect the awareness of saving money rather than CO2 reduction behaviors or behaviors that are good for the environment. The reason "garbage," which was significant in the CFT, was not significant in the PRICE is thought to be that garbage separation is significant because it affects the increase or decrease of CO2 in terms of waste incineration, but garbage separation does not lead to saving behavior, so it was not significant in the PRICE. I have added this to the discussion.

Also, regarding the intercept dummy, in DCE, the main focus is on the interaction effect to see the effect of demographics on attributes. This is in accordance with major studies such as Onozaka & McFaden (2011). The inclusion of intercept dummies in the analysis is very rare, but some studies do use that technique, but it is not mentioned in the conclusion (Yue and Tong, 2009).

p.39, l.618-624,

The reason "not leaving the tap running" and "walking", which were significant in PRICE, were not significant in CFT is that these two behaviors strongly reflect the awareness of saving money rather than CO2 reduction behavior or good behavior for the environment. The reason "separating garbage," which was significant in the CFT, was not significant in the PRICE is thought to be that garbage separation is significant because it affects the increase or decrease of CO2 from the viewpoint of waste incineration, but garbage separation is not significant in the PRICE because it does not lead to saving behavior.

10. if each participant answered 12 choice sets with 3 alternatives each. For example, for a sample size of 104 in NHLEE setting, please explain why did you come up with observations=1248 (table 5 & table 6).

Reply: The number of observations is 1248 for 104 participants x 12 sets. We added this explanation in the footnote in Table 5.

In the footnote in Table 5,

Observations are equal to the # of respondents multiplied by 12 choice sets.

11. Line396: I did not find references [7, 8] applying CC test for the comparison of differences between the estimated mean WTPs.

Reply: References #7 and #8 existed in the reference lists. #7 used the term combinational test and #8 complete combinational test. These are the same. Now they are listed as #6 and #11. We moved that sentence in the first paragraph in 3.3 Hypotheses.

12. If the authors did not provide an opt-out option in their DCE, it implies that participants are forced to buy orange in each round where the min. purchase price is JPY25 per orange. Given the endowment with JPY120 per round and 12 rounds in total (overall endowment JPY1440) in NHLEE setting, please explain how did you come up with an average of JPY1407 (line364) if participants are rewarded based on the sum of the residuals of the endowment minus the price of the selected orange in each round?

Reply: Thank you for pointing this out. I revised Step7 in 2.5.1 NHLEE as follows. This is because of income tax in Japan law.

p.18, l.238-241,

Rewards are calculated by the following equation: show-up fee and the sum of the changes of the endowment minus the price of the selected orange in each round. After a 10% income tax was deducted from the rewards, the remaining money was handed to the participants.

13. Line 413: I did not find the dummy variable of “control” included in the RPL model.

Reply: Thank you for pointing this out. We changed that term Air Conditioning from just Control in Table 3. But it was omitted from Table 6 because it did not reach significance in any of the interaction effects. The results for all variables are in the supplemental material.

14. All studies have limitations. What are the limitations in this study? I suggest to focus on the issues regarding method and study design.

Reply: Thank you for your suggestion. 5.4 Limitations have been added to the Discussion, as follows.

p.40-41, l.634-654

There are three limitations. The first is the sample selection bias between treatments. To compare the laboratory treatments (NHLEE, HLS) and the online treatments (HOL, HOLCT), the responses had to be collected in different ways. In experiments, it is very difficult to recruit hundreds of subjects according to demographic ratios. Therefore, there is a difference in the demographic ratio and sample size between the online survey and the laboratory experiment and survey. For this reason, we tried to improve the bias as much as possible by examining the interaction effect with personal attributes in the analysis. Future tests of the difference in HB should ensure that there is no difference in the subject samples between treatments.

Secondly, as a treatment, we did not do a laboratory survey with CT and non-hypothetical online experiment. If we do these two, we can cover all six treatments: implementation environment × CT presence/absence × monetary incentive presence/absence = 6 treatments. It is relatively easy to communicate CT in the lab. To do a non-hypothetical online experiment is a challenge, as research companies often pay monetary points instead of rewarding with cash, which is against the induced value theory suggested by Vernon Smith.

Third, in the method, we calculated WTP using preference space; since we needed to compare WTP for HB validation, it would be preferable to estimate WTP directly in WTP-space, considering scale effects between treatments. However, we tried to estimate WTP-space from the results of this study using the generalized mixed logit model of NLOGIT, but the estimation was not possible. An appropriate design for estimation in WTP-space is an issue to be explored in future research.

15. Scientific references are missing in the following statements: lines 3-7 / lines 13-15 / lines 16-17 / lines 66-68 / lines 70-81 / lines 76

Reply: We added the references as follows:

lines 3-7: 

Louviere, J.J., Hensher, D.A., and Swait, J.D., Stated choice methods: analysis and applications. 2000: Cambridge University Press.

Hensher, D. A., Rose, J. M., Rose, J. M., & Greene, W. H. (2005). Applied choice analysis: a primer. Cambridge university press.

lines 13-15: 

Aoki, K., Shen, J., and Saijo, T., Consumer reaction to information on food additives: Evidence from an eating experiment and a field survey. Journal of Economic Behavior & Organization, 2010. 73(3): p. 433-438.

Aoki, K., Akai, K., Ujiie, K., Shimmura, T., and Nishino, N., The impact of information on taste ranking and cultivation method on rice types that protect endangered birds in Japan: Non-hypothetical choice experiment with tasting. Food quality and preference, 2019. 75: p. 28-38.

lines 16-17: 

Lusk, J.L. and Schroeder, T.C., Are choice experiments incentive compatible? A test with quality differentiated beef steaks. American Journal of Agricultural Economics, 2004. 86(2): p. 467-482.

lines 66-68: 

UNFCC 2015, The Paris Agreement; Available from

https://unfccc.int/process-and-meetings/the-paris-agreement/the-paris-agreement

lines 70-81: 

METI (Ministry of Economy, Trade and Industry) 2008, Carbon Footprint System in Japan and future direction (In Japanese); Available from

https://www.cfp-japan.jp/common/files/3meti.pdf

Ministry of the Environment 2008, Greenhouse gas emissions and sinks calculation results (In Japanese); Available from

https://www.env.go.jp/earth/ondanka/ghg-mrv/emissions/results-h20.html

Ministry of Economy, Trade and Industry, 2021, Councils and Study Groups: Energy and Environment "Study Group for Pilot Project of "Carbon Offset Products Using CFP” (In Japanese); Available from

https://www.meti.go.jp/committee/kenkyukai/energy_environment.html

line 76: 

SuMPO (Sustainable Management Promotion Organization), 2021, CFP program (In Japanese); Available from

https://www.cfp-japan.jp/

16. Language issue: line 555

Reply: We modified it.

---

## [Decision Letter · Decision Letter 1]

1 Dec 2021

Testing hypothetical bias in a choice experiment: An application to the value of the carbon footprint of mandarin oranges

PONE-D-21-18475R1

Dear Dr. Aoki,

We’re pleased to inform you that your manuscript has been judged scientifically suitable for publication and will be formally accepted for publication once it meets all outstanding technical requirements.

Kind regards,

Luigi Cembalo, PhD

Academic Editor

PLOS ONE

Additional Editor Comments (optional):

Reviewers' comments:

Reviewer's Responses to Questions

**Comments to the Author**

1. If the authors have adequately addressed your comments raised in a previous round of review and you feel that this manuscript is now acceptable for publication, you may indicate that here to bypass the “Comments to the Author” section, enter your conflict of interest statement in the “Confidential to Editor” section, and submit your "Accept" recommendation.

Reviewer #1: All comments have been addressed

Reviewer #2: All comments have been addressed

2. Is the manuscript technically sound, and do the data support the conclusions?

Reviewer #1: Yes

Reviewer #2: Yes

3. Has the statistical analysis been performed appropriately and rigorously? 

Reviewer #1: Yes

Reviewer #2: Yes

4. Have the authors made all data underlying the findings in their manuscript fully available?

Reviewer #1: (No Response)

Reviewer #2: Yes

5. Is the manuscript presented in an intelligible fashion and written in standard English?

Reviewer #1: Yes

Reviewer #2: Yes

6. Review Comments to the Author

Reviewer #1: (No Response)

Reviewer #2: The authors have addressed all of my comments and concerns in the revised manuscript. The paper is now ready for publication.

7. PLOS authors have the option to publish the peer review history of their article (what does this mean?). If published, this will include your full peer review and any attached files.

Reviewer #1: No

Reviewer #2: No

---

## [Editor Report · Acceptance letter]

7 Jan 2022

PONE-D-21-18475R1 

Testing hypothetical bias in a choice experiment: An application to the value of the carbon footprint of mandarin oranges 

Dear Dr. Aoki:

I'm pleased to inform you that your manuscript has been deemed suitable for publication in PLOS ONE. Congratulations! Your manuscript is now with our production department. 

Kind regards, 

on behalf of

Dr. Luigi Cembalo 

Academic Editor

PLOS ONE